# Relationships of temperature and biodiversity with stability of natural aquatic food webs

Qinghua Zhao [1,2,3,4] ✉, Paul J. Van den Brink [1,5], Chi Xu [6], Shaopeng Wang [7], Adam T. Clark[8], Canan Karakoç[9], George Sugihara[10], Claire E. Widdicombe[11], Angus Atkinson [11], Shin-ichiro S. Matsuzaki [12], Ryuichiro Shinohara [12], Shuiqing He[13], Yingying. X. G. Wang[14] & Frederik De Laender [2,3,4]

Temperature and biodiversity changes occur in concert, but their joint effects on ecological stability of natural food webs are unknown. Here, we assess these relationships in 19 planktonic food webs. We estimate stability as structural stability (using the volume contraction rate) and temporal stability (using the temporal variation of species abundances). Warmer temperatures were associated with lower structural and temporal stability, while biodiversity had no consistent effects on either stability property. While species richness was associated with lower structural stability and higher temporal stability, Simpson diversity was associated with higher temporal stability. The responses of structural stability were linked to disproportionate contributions from two trophic groups (predators and consumers), while the responses of temporal stability were linked both to synchrony of all species within the food web and distinctive contributions from three trophic groups (predators, consumers, and producers). Our results suggest that, in natural ecosystems, warmer temperatures can erode ecosystem stability, while biodiversity changes may not have consistent effects.

Whether climate change will increase or decrease the ecological stability of food webs is a fundamental but unresolved question[1,2]. Ecological stability consists of multiple components and combining them enhances our understanding of ecological impacts[3,4]. Structural and temporal stability, which measure stability as the volume contraction rate[5,6] and temporal variation of species abundances[7,8] are two such components, respectively. The volume contraction rate highlights how much population dynamics would change when modifying a parameter (e.g., intrinsic growth rates, species interaction coefficients)[5,6]. When the volume contraction rate is smaller, populations tend to be

[1]Aquatic Ecology and Water Quality Management Group, Wageningen University & Research, P.O. Box 47, 6700 AA Wageningen, The Netherlands. [2]Research Unit of Environmental and Evolutionary Biology (URBE), University of Namur, Namur, Belgium. [3]Institute of Complex Systems (naXys), University of Namur, Namur, Belgium. [4]Institute of Life, Earth and the Environment (ILEE), University of Namur, Namur, Belgium. [5]Wageningen Environmental Research, P.O. Box 47, 6700 AA Wageningen, The Netherlands. [6]School of Life Sciences, Nanjing University, Nanjing 210023, China. [7]Institute of Ecology, College of Urban and Environmental Science, and Key Laboratory for Earth Surface Processes of the Ministry of Education, Peking University, 100871 Beijing, China. [8]Institute of Biology, University of Graz, Holteigasse 6, 8010 Graz, Austria. [9]Department of Biology, Indiana University, 1001 East Third Street, Bloomington, IN 47405, USA. [10]Scripps Institution of Oceanography, University of California-San Diego, La Jolla, CA, USA. [11]Plymouth Marine Laboratory, Prospect Place, The Hoe, Plymouth PL13DH, UK. [12]National Institute for Environmental Studies, Tsukuba, Ibaraki, Japan. [13]Wildlife Ecology and Conservation Group, Wageningen University & Research, Wageningen, The Netherlands. [14]Department of Biological and Environmental Science, University of Jyväskylä, FI-40014 Jyväskylä, Finland. ✉e-mail: qinghua.zhao@unamur.be

more robust to parameter changes, which is indicative of a higher structural stability[5,6]. Temporal stability focuses on temporal variation of species abundance; a smaller temporal variation indicates higher temporal stability[7,8]. At present, these two stability indices are often examined separately, and their joint responses are largely overlooked.

Temperature increases are known to alter ecological parameters of a system such as species interactions and intrinsic growth rates. When, for example, temperature increases consumption rates[9–11], a change of structural stability can ensue, as shown by a recent study on a coastal community of competing species[5]. These findings might not readily translate to multitrophic community types such as food webs, where different trophic levels have different sensitivities to temperature changes[12,13]. To our knowledge, no study examines the effects of temperature on the structural stability of food webs.

Ecological parameter changes following a temperature increase can also lead to effects on species abundance[12,13], and lead to different species showing different degree of variability in population abundance[14,15]. Indeed, temperature can alter temporal stability of food webs, and negative[14], neutral[15], and positive[16] effects have been documented in experimental and simulation studies[14,16,17]. Changes of synchrony of all species within food webs[18,19] can explain these effects, and sometimes there are disproportionate changes in temporal stability of specific trophic groups (e.g., producers, consumers, and predators)[15] that drive these effects. How consistent these mechanisms are across food webs is uncertain, as available studies have focused on single food webs.

Warming goes hand in hand with biodiversity change[20–22], another key factor shaping ecological stability[23–25]. To our knowledge, direct evidence for effects of biodiversity metrics in general on the structural stability in food webs is absent, while higher biodiversity has been shown to have positive[7,26] or neutral[27] effects on temporal stability. How warmer temperatures and biodiversity changes jointly affect ecological stability is uncertain, as most studies only focus on one of these two factors. The co-occurrence of both global changes in natural ecosystems warrants an integrative approach to studying their joint effects[20–22], so as to realistically forecast stability, and thus related functional ecosystem features and services[28–30].

Current conclusions on the effects of warming or biodiversity on ecological stability have mostly been based on short-term experiments or model simulations that consider a limited range of temperature[31,32]. Translating these results to natural ecosystems is challenging. First, species interactions and their responses to temperature can change through time[33,34]. Focusing on short-term effects, therefore, precludes the adaptation of thermal reaction norms and the waxing and waning of species interactions through time[32,35,36], which has been repeatedly observed in natural ecosystems in response to environmental change[34,37,38]. Second, structural stability is traditionally computed from predefined model equations[39], often assuming systems to converge to point equilibria. However, communities in natural ecosystems often exhibit more complex dynamics[24], with species interactions varying with system state, making structural stability a dynamic property[5]. Techniques to study this dynamic behaviour are now available, and their application to field data has allowed studying the effect of environmental variables on structural stability[5,40,41].

In this study, we quantified structural and temporal stability of natural food webs by collating a total of 19 long-term data sets from Europe and North America (seven from freshwater lakes, three from marine and nine from river estuaries), each spanning 10 to nearly 30 years of data. First, we applied empirical dynamic modelling (EDM) to the time series of the 19 planktonic food webs to infer trophic interactions among consumers and resources from recorded population dynamics, thereby reconstructing the interaction networks. Next, we estimated time-specific net species interactions by using the multiview distance regularised S-map, as such reconstructing the dynamics of the Jacobian matrix, showing how structural stability changes through time (Fig. 1). Temporal stability in each of the 19 food webs was estimated as the coefficient of variation of total community abundance. Finally, we examined the relationships between temperature/biodiversity and the two stability indices (structural and temporal stability). We found that higher temperatures were associated with lower structural and temporal stability, while biodiversity indices had no consistent associations. We interpret these associations as evidence of temperature and biodiversity effects, and we use both terms ('associations' and 'effects') hereafter to represent the associations. While we acknowledge the correlational nature of temperature and biodiversity effects on stability in our study, we believe that our interpretation is supported by existing knowledge on temperature and biodiversity effects and the nature of our study design (Fig. 1). Finally, different trophic groups (predators, consumers, or producers) had different contributions to structural and temporal stability. Synchrony of all species within the food web had a positive effect on the food web's temporal stability.

## Results

We first quantified the time-varying Jacobian matrix of each food web with the multiview distance regularised S-map[41], from which the structural stability of each food web was measured as the volume contraction rate (VCR), which is the divergence of a vector field and is equivalent to the trace (Tr(J)) of the Jacobian matrix[5,6] (see methods). Smaller values of Tr(J) (i.e., VCR) indicate lower sensitivity to parameter perturbations[6], i.e., a higher structural stability[5,6]. Next, we computed temporal stability of each food web as the coefficient of variation of total community abundance, by using a time window of 1.5 years. A larger coefficient of variation indicates lower temporal stability.

Across food webs, we found that temperature was consistently associated with lower structural and temporal stability of food webs, as temperature increases resulted in a higher Tr(J) and a higher coefficient of variation (Fig. 2a, d). In contrast, species richness was associated with lower and higher structural and temporal stability, respectively (Fig. 2b, e). Simpson diversity was only associated with higher temporal stability (Fig. 2c, f). These trends were robust to changing the length of the time window to compute temporal stability of food webs (Fig. S1), and to the inclusion of rare species—which are normally excluded from similar analyses (Fig. S2). Within food webs, temperature and biodiversity effects on structural and temporal stability were system dependent (Fig. 2). In 13 (11) out of 19 food webs, temperature had negative effects on structural (temporal) stability (Fig. 2a, d). Similarly, in 14 (17) out of 19 food webs, species richness had negative (positive) effects on structural (temporal) stability (Fig. 2b, e). Simpson diversity in 13 out of 19 food webs had positive effects on temporal stability (Fig. 2f).

Structural stability of food webs did not vary along latitudes, while temporal stability was higher at higher latitudes (Fig. S3). Finally, we found that temperature had no effect on either biodiversity index (Fig. S4). Thus, warmer temperatures mainly reduced stability directly, and less so indirectly by changing biodiversity (e.g., temperature → biodiversity → stability).

The effects of temperature on structural stability of food webs were mostly driven by temperature effects on the contribution from predators. This contribution, which is the sum of those diagonal elements of Jacobian matrix J that belongs to predators and includes the aggregated effects of other species on predator species, increased (Fig. 3a), thus increasing Tr(J) and therefore decreasing structural stability. We did not find temperature effects on contributions from consumers (Fig. 3d) or producers (Fig. S5a). Species richness increased the contributions from consumers (Fig. 3e), while we found species richness had no effects on the contributions from predators (Fig. 3b) or producers (Fig. S5b).

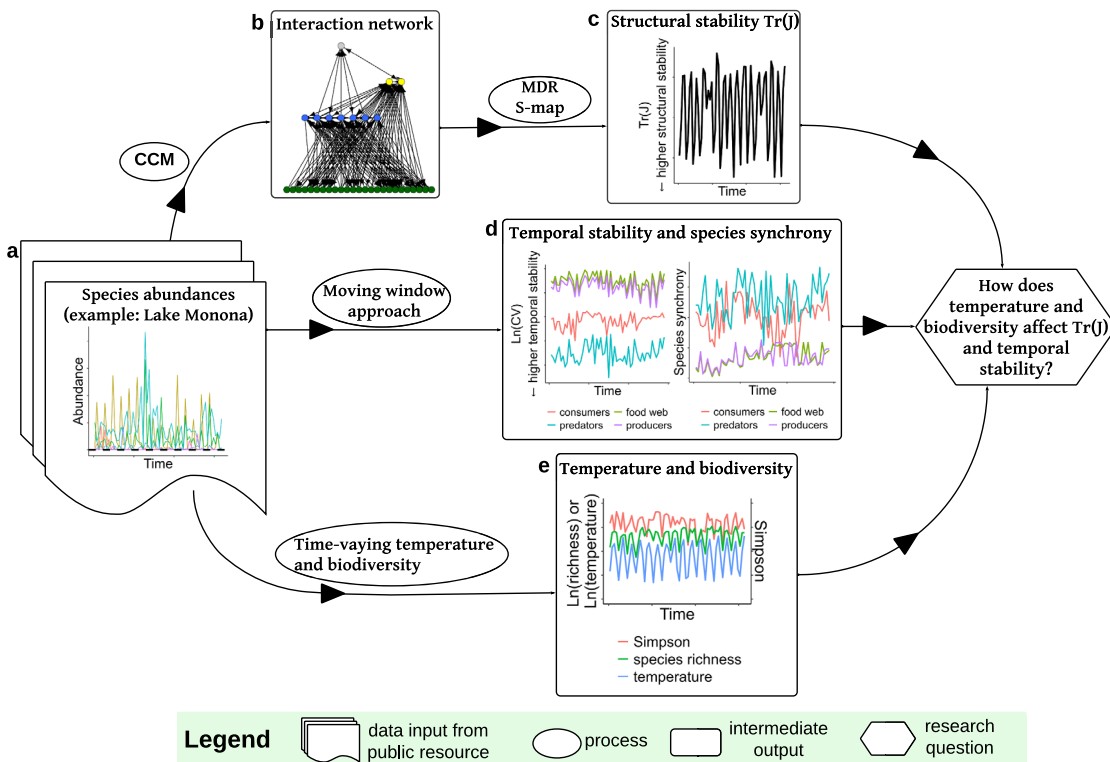

**Fig. 1 | An overview of the data and methods used in this paper, illustrated with the data from Lake Monona. a** Species abundances over time are shown, where the horizontal black dashed line indicates a zero species abundance (species absence). **b** We reconstruct the interaction network using convergent cross mapping (CCM). **c** We compute structural stability for each time point as the trace of the local Jacobian matrix Tr(J). The Jacobian matrices are inferred by multiview distance regularised S-map (MDR S-map). The size of the Jacobian is fixed over time, as $s \times s$ (row × column), where $s$ is the number of network nodes in the food web. **d** The temporal stability of the food web and of each trophic group is computed as the coefficient of variation (CV) of the total community abundance and each trophic group, respectively, using a time window of 1.5 years. Species asynchrony of the food web and of each trophic group is computed using the moving window. **e** Time-varying species richness (i.e., the sampled species richness based purely on species presence and absence) and Simpson diversity are calculated from the data, and combined with time-varying temperature data.

Effects of warmer temperatures and biodiversity on temporal stability of food webs were related to the synchrony of all species within the food web (Fig. 3g–i). Warmer temperature increased this synchrony (Fig. 3g), while species richness and Simpson diversity decreased it (Fig. 3h–i). The increase or decrease of this synchrony was associated with lower or higher temporal stability of food webs, respectively (Fig. S5d). Furthermore, we found that temperature and species richness mostly affected the temporal stability of producers (Fig. S6a–b), which then altered the temporal stability of the whole food web (Fig. S7a). Conversely, Simpson diversity comparably increased the temporal stability of all trophic groups (producers, consumers, predators) (Fig. S6c, f, i), which then increased the temporal stability of the whole food web (S7a-c). In addition, the contrasting effects of temperature and biodiversity on the temporal stability of trophic groups (Fig. S6a-c, f, i) were also related to contrasting effects on species synchrony of corresponding trophic groups (Fig S8a–c, f, i–l). Moreover, the synchrony of producers determined the synchrony of the whole food web more than did consumers or predators (Fig S7d-f).

Finally, temperature and biodiversity also altered specific structural aspects (i.e., link density $L/S$, with number of links $L$ and number of network nodes $S$), which in turn affected temporal food web stability (Fig. S9a, b, d). Higher mean temperature was associated with a lower link density, which reduced temporal stability of food webs (Fig S9a, d). In contrast, higher mean species richness was related to higher link density, which then increased temporal stability of food webs (Fig S9b, d). We did not find that other structural aspects (i.e., connectance $L/S^2$ and food chain length) had effects on the temporal or structural stability of food webs (Fig S10).

## Discussion
Our results show that warmer temperatures are associated with lower structural and temporal stability, while we found biodiversity had no consistent effects on either stability property. The contributions from predators and consumers, the synchrony of all species within the food web, and temporal stability of different trophic groups explain these results.

Temperature has been found a strong driver of structural stability in a competitive coastal community[5]; here, we show that temperature reduces structural stability across 19 planktonic food webs. Within food webs, temperature effects on structural stability were system dependent, albeit mostly negative. Lower structural stability in warmer temperatures indicates a lower robustness to parameters perturbations. Temperature effects on those diagonal entries of the Jacobian matrix that belong to predators are possible explanations to the decrease of structural stability we found (Fig. 3a). The biological mechanism explaining the temperature effect on the diagonal entries for predators could involve effects on density-dependent (e.g., consumption rates and self-limitation) or density-independent contributions (e.g., intrinsic growth rates, generally negative for predators) to the per-capita growth rate of predators (see Supplementary Information. Part 1). Increases of the per-capita growth rate of predators could be attributed to 1) increases of predators' consumption rates, or 2) decreases of predators' intrinsic growth rates, or 3) decreases of predators' self-limitation, or 4) increases of predators' consumption rates

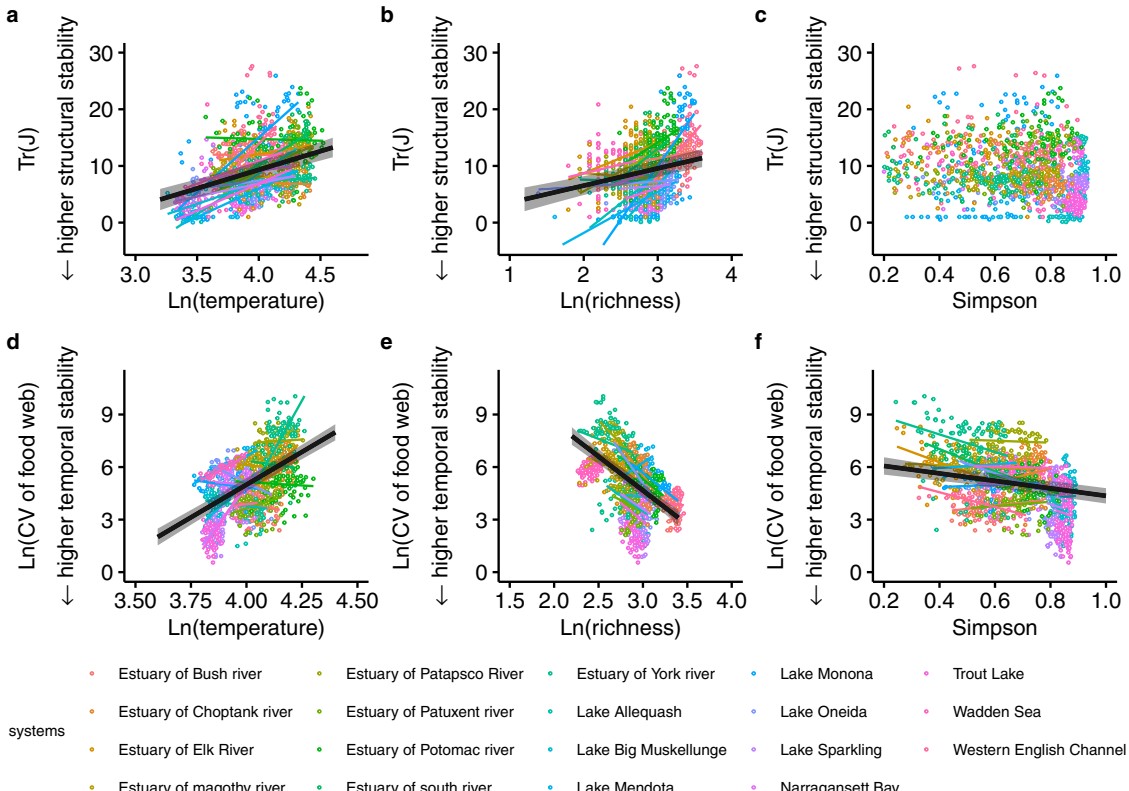

**Fig. 2 | The effect of temperature, species richness, and Simpson diversity on structural stability and temporal stability of food webs. a–c** structural stability (Tr(J)) of food webs. **d–f** temporal stability (coefficient of variation of species abundance, CV) of food webs. The CV is natural log-transformed. A smaller Tr(J) or smaller CV indicates higher stability. **a–c** Coloured points correspond to values of each food web in each season of each year (total points, $n = 1572$ biologically independent samples). **d–f** Coloured points ($n = 1477$) correspond to values of each food web in each moving window (window width = 1.5 years). **a–f** The bold black lines and error bands depict the significant best-fit trendline and the 95% confidence interval in the linear mixed model (two-sided) across 19 food webs, respectively. The non-bold coloured lines indicate the best-fit trendline in the linear models (two-sided) within each food web. For statistical results see table S1.

that are greater than the increases of the other parameters 2)–3). Numerous studies have shown that temperature can increase predator consumption rates[9,42–44], decrease self-limitation[45], or increase predator consumption rates more than it does other parameters[46,47]. For example, Lang et al. (2012)[48] have shown that warming increased consumption by a predacious ground beetle to a greater extent than it increased energy losses. Simulation studies have further shown that such parameter changes can drive predator-consumer systems from stable equilibria to a limit cycle[46,49] or chaos[50,51].

We found that species richness decreased structural stability. A similar result is found when adopting an alternative approach to compute structural stability in ecological communities[52]: the dimensionality of parameter space grows as more species are added, which reduces the fraction of that space leading to positive population densities. In addition, the negative effects of species richness on structural stability found in this study (Fig. 2b) can be explained by positive effects of species richness on the contribution from consumers (Fig. 3e). The mechanism behind this result could be again attributed to changes of growth and consumption rates, and self-limitation (see Supplementary Information. Part 2). In contrast, Simpson diversity did not affect structural stability and the diagonal entries of any trophic group (Figs. 2c, 3c, 3f; Fig S5c).

Previous experimental and simulation studies have shown that warmer temperatures can lead to lower temporal food web stability[14,16,17]. Here we show that this finding extends to natural food webs, analysing multiple long-term data sets. Lower temporal stability in warmer temperatures indicates a greater degree of variability in species abundance with respect to its mean. Our finding that the synchrony of all species within the food web, temporal stability of producers, and link density underpins this result, supports previous empirical findings[15,18,19,53–56]. Warmer temperature was linked to higher synchrony of all species within the food web, which might be caused by warmer temperature increasing consumer-producer interactions, tightening the control of consumers on producers and resulting in synchronous changes in abundance dynamics[57,58]. In addition, higher mean temperature was associated with a lower link density, which could be expected when higher temperature reduces the number of links by favouring predators to be specialists rather than generalists[56]. Higher temperature favouring specialisation in predators is supported by evidence that increasing temperature can increase predator attack rates more than it decreases handling time by altering activation energy in Arrhenius function[56,59].

In contrast to temperature, we found that biodiversity reduced the synchrony of all species within the food web, but increased temporal stability of trophic groups and link density, which therefore increased temporal food web stability, again supporting previous experiments, field studies, and theory[7,26,60–62]. Finally, species synchrony (and temporal stability) of producers determined the synchrony (and temporal stability) of the whole food web more than did consumers or predators, supporting recent findings established in short-term experiments[15].

Natural food webs are inevitably under-sampled. Most notably, our study focuses on planktonic species, reproducing on time scales of days. Larger (e.g., fish, mammals) and smaller biota (e.g., bacteria) are excluded from our analyses because they were not reported, or only measured infrequently. We acknowledge that

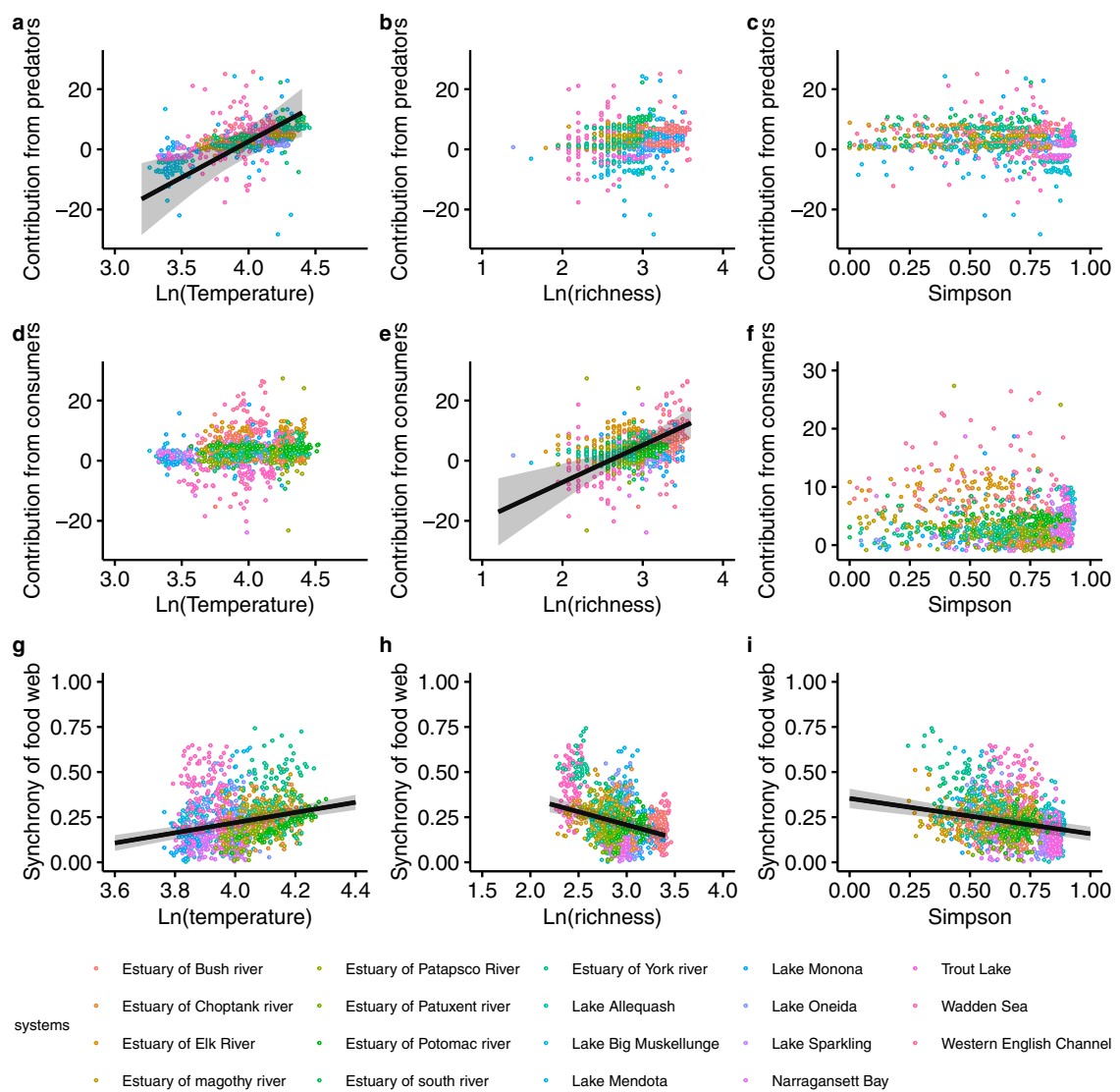

**Fig. 3 | The effect of temperature, species richness and Simpson diversity on the contribution from predators and consumers to structural stability, and on the synchrony of all species within the food web. a–f** Coloured points correspond to values of each food web in each season of each year (total points, $n = 1572$ biologically independent samples). **g–i** Coloured points ($n = 1477$) correspond to values of each food web in each moving window (window width = 1.5 years). **a–i** The bold black lines and error bands depict the significant best-fit trendline and the 95% confidence interval in the linear mixed model (two-sided) across 19 food webs, respectively. For statistical results see table S1.

these biota can play critical roles in ecosystem functioning and can mediate planktonic species dynamics[63]. Because their abundance could not be accounted for in a consistent way, their contribution in this study is only implicitly present. Given that some studies showed warmer temperatures can reduce the synchrony between fish and lower trophic planktonic species[64,65], we expect that the explicit inclusion of fish in this analysis might weaken the negative effect of temperature on temporal stability.

We have applied time series analysis to quantify the effects of temperature and biodiversity on two types of stability in complex food webs. We found that warmer temperatures, in natural ecosystems, were associated with lower structural and temporal stability, while biodiversity had no consistent effect on stability. This suggests that the methods assuming ecosystems to exhibit static equilibria may not be sufficient to evaluate how global change affects the stability of networks. Given the increasing amount of available data from natural ecosystems[66–68], our work paves the way for the application of long-term monitoring data sets to investigate the effects of (a)biotic factors on ecosystems' structure, function, and stability.

## Methods

### Time series data of food webs

We used 19 long-term time series data sets representing 7 freshwater ecosystems (lakes), 3 marine ecosystems (Western English Channel, Wadden sea and Narragansett Bay), and 9 rivers (estuary of rivers) to test the responses of food web stability to biodiversity and temperature (Table S2). Of those, 18 data sets were obtained from the publicly available open database from North Temperate Lakes (NTL-LTER)[69,70], Data Observation Network for Earth[71], Center for Global Environmental Research[72], UK Centre for Ecology & Hydrology[73], Pangaea[74] and Waikato Regional Council[75]. The last dataset from the Western English Channel is archived at the British Oceanographic Data Centre www.bodc.ac.uk and was obtained upon request from Plymouth Marine Laboratory. We selected the data sets using the following search criteria: (1) the number of trophic levels was at least 2, (2) taxa were identified to species level or to finest taxonomical level as possible (generally species level), and (3) temperature was available. Next, we removed the data sets that were only sparsely sampled (e.g., yearly or semi-annually). We only kept data sets with seasonal sampling of all

variables (if multiple samplings were conducted per season, e.g., monthly and bimonthly sampling data sets, those were averaged). Here, the seasonal resolution (trimonthly) is the shared consensus that can be applied to all 19 data sets, and the seasonal average here is also the most representative measure across all data sets, because the equal sampling interval is necessary for comparison across systems and also for EDM analysis[76]. After the seasonal average, there were two missing points across the whole dataset (accounting for 0.17% of whole data). One missing point was from Narragansett Bay (winter of 1979), and the other one was from Wadden Sea (summer of 1989). Those 2 missing data points were linearly interpolated using na.approx function in the package of zoo[77]. Then, the abundance of each species across all data sets was scaled to the same unit (individuals per litre). Note that fish species were excluded from analysis, because they were either not reported or yearly sampled only, and because the unit (catch per unit effort) of fish species changed over sampling time. We thus found 19 long-term seasonal data sets consisting mainly of plankton species and spanning from 10 to 30 years and originating from the continents of North America and Europe (Table S2). Recent work showed that the plankton species from natural ecosystems had a greater proportion of chaotic time series than others (e.g., fishes)[78], which indicated that the plankton species could fit nicely for EDM analysis. Then, we divided all species into producers, herbivores, omnivores, and predators by their diet[79,80]. Next, we retained the species which were encountered at least once per 1.5 years (at least 1 nonzero abundance data out of 6 data points) for the analyses, to exclude the low-frequency species which include too many zero values in their time series. Too many zero values is a general statistical problem in time series analysis[76]. Before EDM analysis, all time series data of abundance and temperature were normalised to have zero mean and unit variance, while raw data was used to compute temporal stability because the mean and variance are parts of the equation to compute temporal stability.

## Inferring causal interactions among species

For each dataset, we identified the causal links (e.g., competition and predation) across all potential species pairs in each dataset using convergent cross mapping (CCM)[40] (see three brief videos for a summary introduction link1: http://tinyurl.com/EDM-intro). CCM is based on Takens's theorem, which proves that as a generic property it is possible to construct a shadow version of the original attractor of a dynamical system by substituting time lags of the observable variables for the unknown variables[40,81]. An important consequence of this is that if the causal variable X and affected variable Y belong to the same dynamical system, information on the causal variable X is encoded in the affected variable Y. Thus one can predict the states of causal variable X using the affected variable Y. CCM infers causality by measuring the extent to which the causal variable X has left signatures in the time series of the affect variable Y; a procedure known as cross-mapping (cross-prediction)[40]. In this study, the appropriate embedding dimensions for cross-mapping $E$ were determined by univariate simplex projection[82], examining values of $E$ from 2 to square root of $n$, where $n$ is the length of the time series[77]. Following Deyle et al.[83], we examined the same range of $E$ across all studied data sets. Thus, we computed $n$ as the geometric mean time series length across all data sets. $E$ was finally examined from 2 to 9 across all data sets. We used simplex projection to select the best $E$ that gave the lowest mean absolute error[34,77].

The first necessary criterion to test for causality among variables requires that cross-mapping skill in the real time series need to be higher than the ones in null surrogates–generated time series containing associations or temporal patterns that are conservatively asserted to be non-causal. Because our data sets come from field seasonally-monitoring, following Deyle et al.[83], we used null surrogates designed to factor out seasonality as a contributing factor in CCM. Specifically, for any causal variable X, we calculated the yearly averages

of X and seasonal anomalies as the difference between the observed X and this yearly average. Then, we randomly shuffled the seasonal anomalies and added them back to the yearly averages to generate surrogates with randomised time dependence between anomalies. Thus, the new surrogate time series ($X_{sur}$) have the same seasonal average as X, but with randomly shuffled anomalies. The conservative reasoning described by Deyle et al.[83,84] is that if X indeed causes Y in a manner that extends beyond the effects of seasonality, then Y should be sensitive not only to the seasonal components of X, but also to the anomalies; thus Y should cross predict the real time series X better than the surrogate time series $X_{sur}$[83]. The analyses in this study are based on generating 100 null seasonal surrogates for X[85].

The second necessary criterion for testing causality is the convergence towards higher cross-mapping skill as library length (i.e., the number of points used for state space reconstruction) increases[40]. Because longer library length increases the density of points in the reconstructed attractor, the nearest neighbouring points used to make predictions from an attractor will be more accurately determined[82], which in turn leads to improved predictions[40]. Convergence can be examined by testing whether there is a significant monotonic increasing trend in cross-mapping skill ρ with an increase of library length by Kendall's τ test[86], and whether ρ at the largest library length is significantly higher than the one at the smallest library length (ρ) by Fisher's Z test[86]. In this study, library length was set from the smallest (10) to the largest length (i.e., the length of the entire time series).

Throughout this paper, an interaction link (e.g., X→Y) was regarded as causal if both of the two criteria above were satisfied: (1) predictive skill ρ in the real-time series was higher than the 95% confidence intervals of surrogates[34]; (2) both Kendall's τ test and Fisher's Z test were statistically significant ($P < 0.05$) for testing convergence[86]. As additional consideration to (1) above, to accommodate the fact that the causal variables (e.g., prey species) can exhibit time-delayed effects on the affected variables (e.g., predator species)[87], we carried out 0 to 6 month (0-2 time point) lagged CCM analyses[88], in which we retained the CCM with the time lag resulting in the highest ρ[89]. Furthermore, because causality is transitive and can occur indirectly through a transitive causal chain[40] and to narrow our focus on direct linkages, we used the 0 to 6 month (0-2 time point) lagged CCM analyses to detect and remove the suspected indirect link[90]. Briefly, if the variable X unidirectionally causes Y (X→Y, e.g., producers→herbivores) and then the variable Y unidirectionally causes Z (Y→Z, e.g., herbivores→predators), an indirect causality (X⇢Z, e.g., producers⇢predators) may thus emerge if the effect of X on Y is sufficiently strong[40,90]. The indirect link (X⇢Z) is detected and removed when it has both: (1) a larger negative time lag, and (2) a lower predictive skill ρ than the direct link (X→Y), due to transitivity[40].

Overall, we found the number of links per reconstructed interaction network was between 91 (Estuary of Magothy river) and 557 (Western English Channel) links, with an average of 207 links per food web (Fig. S11–S12). Link density (i.e., $L/S$, with $L$ the number of links and $S$ the number of network nodes[91], Fig. S11–S12) was between 4.55 (Estuary of Magothy River) and 13.92 (Western English Channel), and on average 7.91. Foodweb connectance[61] (i.e., $L/S^2$) was between 0.21 (Narragansett Bay) and 0.68 (Wadden Sea), and 0.32 on average.

## Quantifying time-varying interaction strength among species

Once the causal links among variables were established, we attempted to quantify the time-varying strength and direction of effects among causal variables using the multiview distance regularised S-map (MDR S-map)[41]. Here, the regression coefficients approximate the interaction strength in the discrete-time Jacobian matrix $\frac{\partial x_i(t+1)}{\partial x_j(t)}$[41]. $x_i(t+1)$ is the abundance of species $i$ at time point $t+1$ and $x_j(t)$ is species $j$ abundance at time $t$. The MDR S-map was used here because it had a higher accuracy to recover Jacobian matrix than other techiques[41], and

because the embedding dimension $E$ in this study was smaller than the number of species (causal variables) in each food web (Fig. S11–S12).

The MDR S-map is a nonparametric method to reconstruct high-dimensional time-varying interaction networks for complex systems[41]. It works by linking two methods (multiview embedding and regularised S-map)[92,93], but shows a higher accuracy than each one alone[41]. The MDR S-map consists of two steps. The first step is to recover the neighbourhood relationships among high-dimensional data points from numerous low-dimensional state space reconstructions (multiview SSR). Then, one computes Euclidean distances between data points under the optimal embedding dimension $E$. Next, one collects these distances to achieve the multiview distances $d^E$ and to further compute the data weights $\mathbf{W}_t^E$. For the second step, the data weights $\mathbf{W}_t^E$ from first step and regularisation are incorporated into S-map to estimate high-dimensional interaction strength. Specifically, the formula that calculates interaction strength ($\mathbf{B}_t$) is described as

$$\hat{\mathbf{B}}_t = \arg\min_{\mathbf{B}_t} \left( \left\| \sqrt{\mathbf{W}_t^E}(\mathbf{X}_{t+1} - \mathbf{X}_t\mathbf{B}_t) \right\|_2^2 + \lambda[\alpha|\mathbf{B}_t|_2^2 + (1-\alpha)|\mathbf{B}_t|_1] \right) \quad (1)$$

where $\mathbf{W}_t^E$ is the local weight matrix, which is the weight obtained from the exponential decay function of Euclidean distance, with $\mathbf{W}_t^E = \exp(-\frac{\theta d^E}{\mathrm{mean}(d^E)})$. $\theta$ is a state-dependency (nonlinearity) parameter.

$d^E$ contains the multiview distances, which depict the neighbours of a high-dimensional system state via an ensemble of numerous distances measured in low-dimensional state space reconstructions (SSR) at the optimal embedding dimension ($E$). The optimal $E$ for a network node was determined by univariate simplex projection[82]. The multiview distances $d^E$ are calculated as the weighted average among all Euclidean distances (i.e., $d(X^c(t_\mu), X^c(t_\nu))$) between every pair of embedded states observed at various time points for a network node, with $d^E = \sum_{\forall c} w_c d(X^c(t_\mu), X^c(t_\nu))$. $w_c$ is proportional to the forecast skill $\rho$ and $\sum_{\forall c} w_c = 1$. The c denotes any combination consisting of causal variables for a target network node in SSR. In practice, there are too many combinations because of network dimensionality $m$ is larger than $E$ ($m > E$). Thus, in this study, we randomly generated 1000 SSR from combinations of causal variables and a target network node, and finally kept the top 100 SSR with the highest forecast skills regarding the target network node, with the consideration of computational efficiency[41]. In addition, $\lambda$ is the penalised factor, and $\alpha$ is the adjusted parameter to balance the regularisation. Thus, the solution of interaction strength $\mathbf{B}_t$ in the MDR S-map algorithm depends on state-dependent parameter $\theta$, the penalised factor $\lambda$, and the adjusted parameter $\alpha$. The elements in $\mathbf{B}_t$ at Eq. (1) approximate interaction strengths among species $\frac{\partial x_i(t+1)}{\partial x_j(t)}$. Finally, the best parameter combination ($\theta$, $\lambda$, $\alpha$) for each network node and estimated interaction strengths are the ones that minimise rMSE of the one-step forecast on the target network node in $t+1$, based on cross-validation[41].

### Computing structural and temporal stability

We first computed the time-varying structural stability, considering non-equilibrium time series generated by nonlinear dynamical systems:

$$\dot{X} = \mathbf{f}(\mathbf{X}, \boldsymbol{\eta}) \quad (2)$$

Here, $\mathbf{f}$ is an unspecified vector field (or dynamic model). $\mathbf{X}$ is a vector of state variables (i.e., species abundance). $\boldsymbol{\eta}$ is a vector of environment-dependent parameters (e.g., rates of interactions such as consumption rate). Structural stability is measured as volume contraction rate VCR[5,6], which is the divergence of a vector field and equivalent to trace Tr(J) of the Jacobian matrix ($\nabla \cdot \mathbf{f}(\mathbf{X}, \boldsymbol{\eta}) = \mathrm{Tr}(\mathrm{J})$)[5,6]. Smaller values of Tr(J) (i.e., VCR) indicates lower sensitivity to parameter perturbations, i.e., higher structural stability[5,6]. Based on the Jacobian matrices calculated by multiview distance regularised S-map at each time point (see the previous section), structural stability (i.e., Tr(J)) was directly computed as the trace of the Jacobian matrices at each timestep[5,6].

Second, we computed the temporal stability as the coefficient of variation of whole community abundance CV (temporal variation). CV was calculated using a moving window with 1.5 years (window width = 6-time points) for species abundance of each food web. A smaller CV indicates larger temporal stability. We changed the width of the time window to 3 years (12-time points), and our results were robust (Fig. S1).

Finally, we computed the temporal stability of each trophic group (i.e., producers, consumers, and predators), by using a moving window with 1.5 years (window width = 6-time points). Specifically, the temporal stability of producers was computed as CV of producer's population abundance. Similarly, the temporal stability of consumers was computed as CV of primary consumer's population abundance. Temporal stability of predators was calculated as the CV of predator's population abundance (i.e., including the omnivores, secondary and higher consumer's trophic level).

### Quantifying the effects of temperature and biodiversity on stability

Before quantifying these effects, we computed biodiversity (i.e., Simpson diversity and species richness) over time. Simpson diversity on each time point was computed as $1 - \sum p_i^2$, where $p_i$ is the proportional abundance of species $i$. Species richness in each time point was computed as the sampled species richness, i.e., the number of species that was observed to have positive values of abundance at that time point. Sampled species richness is useful as it reflects changes in the underlying relative abundances of species (historically referred to as community structure[94–96]). We also considered the Shannon diversity index ($-\sum p_i \ln(p_i)$), but it exhibited high correlations with Simpson diversity for all 19 data sets (0.91-0.98 in Pearson's correlation, Table S3) and was thus omitted.

Next, we applied linear mixed models to test for the effect of temperature, species richness, and Simpson diversity on structural stability. We treated year, sampling locations (e.g., lake Mendota and Trout Lake), and season as random factors to exclude the potential confounding effects of them. We adopted the same approach for the analysis of temporal stability. Specifically, we applied linear mixed models to test for the effect of temperature, species richness, and Simpson diversity on temporal stability of whole community and each trophic group, but only treated the sampling locations as a random effect. Because temporal stability was calculated using a moving window, the year and season were factored out.

For each of the two stability indices, temperature and species richness were natural log transformed before analysis. Given that values of temperatures were negative in winter, we transferred temperature in all 19 sites from Celsius to Fahrenheit (°C to °F), before natural log transformation. Temporal stability was also natural log transformed before analysis to minimise the variance across sampling sites.

### Quantifying the contribution of trophic groups on structural stability

Since structural stability Tr(J) for each food web was computed as the sum of each diagonal element; that is, sum across all trophic levels (i.e., producers, consumers, or predators); the effects of temperature/biodiversity on structural stability can be understood how the diagonal elements from each trophic groups were changed by the two effects. Specifically, the contribution of Tr(J) by producers, including the aggregated effect of other species on producer's species (hereafter

called "contribution from producers") was thus computed as the sum of these diagonal elements that only belonged to producers. Similar computations for the contribution of Tr(J) by primary consumer's trophic group on structural stability ("contribution from consumers"), and contribution of Tr(J) by omnivores, secondary and higher consumer's trophic level on stability ("contribution from predators").

Next, we applied linear mixed models to test for the effect of temperature, species richness and Simpson diversity on the contribution from producers, by treating the year, sampling locations (e.g., lake Mendota and Trout Lake), and season as random effects to exclude the potential confounding effects. We adopted the same approach for the analysis of the contribution from consumers and the contribution from predators.

### Quantifying the contribution of synchrony on temporal stability of whole community

The degree of synchrony φ of all species within the food web was quantified as

$$\varphi = \sigma^2 / \left( \sum_{i=1}^{s} \sigma_i \right)^2 \tag{3}$$

$\sigma^2$ is the variance of whole community abundance and $\sigma_i$ is the s.d. of abundance of species $i$ in a food web with s species. Species synchrony $\varphi$ is ranging from 0 (perfectly asynchronized of species fluctuations) to 1 (perfectly synchronised of species fluctuations)[15,97,98]. We did the same computation for species synchrony of each trophic group (producers, consumers, or predators).

Then, we applied linear mixed models to test for the effect of synchrony of all species within the food web on temporal stability of food web, by treating sampling locations (e.g., lake Mendota and Trout Lake) as random effects. Similarly, linear mixed models were employed to test for effects of species synchrony of each trophic group on temporal stability of each trophic group, by treating sampling locations as random effects. Finally, we applied linear mixed models to test for the effects of temperature, species richness and Simpson diversity on the synchrony of all species within the food web, or synchrony within each trophic group, again treating sampling locations as random effects.

### Quantifying the effects of temperature on biodiversity
Across 19 food webs, we applied linear mixed models to test for the effect of temperature on either species richness or Simpson diversity, by treating year, sampling locations (e.g., lake Mendota and Trout Lake), and season as random effects. By doing so, one can infer the indirect effects of temperature on structural (or temporal) stability, via direct temperature effects on biodiversity.

### Sensitivity analysis
Given that previous studies showed that some of low-frequency rare species may contribute to stability patterns[99,100], we extended the analysis to include a large number of rare species; these were species which appeared at least once per 2 years (1 nonzero abundance data out of 2 years). The inclusion of rare species in the analysis did not change the conclusions (Fig. S2).

The CCM and S-map were performed using "rEDM" package (https://cran.r-project.org/src/contrib/Archive/rEDM, version 1.2.3). The multiview distance regularised S-map was performed following Chang et al. (2021)[41]. Linear mixed modelling was performed using the lme4 package (https://cran.r-project.org/src/contrib/Archive/lme4, version 1.1.27.1). The missing data points after seasonal average were linearly interpolated using the zoo package (https://cran.r-project.org/src/contrib/Archive/zoo, version 1.8-11). All statistical analyses were performed in R 4.1.2[101].

### Reporting summary
Further information on research design is available in the Nature Portfolio Reporting Summary linked to this article.

## Data availability
The raw data used in this study are publicly available (see Methods and Table S2). Station L4 marine data (Western English Channel) are archived and available from the British Oceanographic Data Centre BODC (www.bodc.ac.uk) with the most recent versions and are freely available upon request to Dr. Claire Widdicombe (clst@pml.ac.uk) and Dr. Angus Atkinson (aat@pml.ac.uk) at Plymouth Marine Laboratory. The structural and temporal stability data generated in this study have been deposited in the Zenodo database (https://doi.org/10.5281/zenodo.7877806)[102]. These data are also available on GitHub (https://github.com/QZhao16/aquatic.foodweb.stability).

## Code availability
Programming code for empirical dynamic modelling (EDM) to infer causal links among species, interaction networks, and structural stability of the food web is available on GitHub. (https://github.com/QZhao16/aquatic.foodweb.stability), and on Zenodo. (https://doi.org/10.5281/zenodo.7877806)[102].

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

## Acknowledgements
We thank Chun-Wei Chang and Chih-hao Hsieh for their guidance in using multiview distance regularised S-map. We thank Ethan R. Deyle who assisted us with the S-map technique. We thank all participants in each long-term monitoring site. We thank members of the Lab of Environmental Change and Community Ecology at the University of Namur for their discussions. F.D.L. acknowledges funding from the concerted research action (ARC) from the special research fund (Convention 18/23-095). Q.H.Z. is supported by the China Scholarship Council (No. 201606190229). S.P.W. is supported by the National

Natural Science Foundation of China (31988102). The Western English Channel time series is funded by the UK Natural Environment Research Council through its National Capability Long-term Single Centre Science Programme, Climate Linked Atlantic Sector Science, grant number NE/R015953/1. C.X. is supported by the National Natural Science Foundation of China (32061143014). G.S. acknowledges the National Science Foundation (DEB-1655203 and ABI-1667584), the Department of Interior (NPS-P20AC00527), the McQuown Fund and the McQuown Chair in Natural Sciences, University of California, San Diego.

## Author contributions

Q.H.Z., F.D.L., A.T.C., C.K., G.S., S.P.W., and C.X. designed the research. Q.H.Z. performed the study and analysed the data. Q.H.Z, F.D.L., A.T.C., C.K., and G.S. contributed to EDM analysis. Q.H.Z. and F.D.L. led the writing, with contributions from P.J.V.d.B., C.X., S.P.W., A.T.C., C.K., G.S., C.E.W., A.A., S.M., R.S., S.Q.H. and Y.X.G.W.

## Competing interests

The authors declare no competing interests.
