## [Peer Review File · Nature Communications]

Reviewer comments, first round

Reviewer #1 (Remarks to the Author):

The authors apply empirical dynamic modeling (EDM) to determine effects of temperature and biodiversity on the structural stability and temporal stability of natural aquatic systems. They take advantage of the fact that there are long-term time series from 19 aquatic food webs, including lakes, rivers and coastal marine sites. The authors use convergent cross mapping to infer species interactions and S-maps to estimate interaction strengths through time. This allows both temporal and structural stability to be estimated from the constructed Jacobian, where recent theory, volume contraction rate, is used to estimate the structural stability from the trace of the Jacobian.

Comments

Lines 86-90. "First, temperature dynamics in nature include diurnal, seasonal, and annual fluctuations and span broad temperature gradients, which can influence ecological stability in a diversity of ways that may not be feasibly studied in experimental approaches." I don't agree here. Mesocosm experiments can be performed in which temperature can be varied in almost any desired way to simulate seasonal variation, etc. In fact, there is an advantage in such experiments, which can exclude many possible confounding factors, such as nutrient concentrations.

Lines 136-137. "Simpson diversity was only associated with higher structural and temporal stability (Figure 1c,f)." Lines 158-159. "Simpson diversity decreased with synchrony (Fig. 2h-i)" Line 166 "but Simpson diversity was associated with higher temporal stability of producers, consumers, and predators (Extended Data Fig. 2c, 2f, 2i)". The authors need to state exactly which version of Simpson's diversity index is used. As far as I can see, the authors do not show the formula they used. The basic form of Simpson's diversity index is D . Diversity decreases with increases in Simpson's index, D . Diversity is zero when Simpson's index is 1. If that is what the authors used, then, my reading from the figures is that temporal stability decreased with diversity (according to Simpson's index) (Fig. 1f), diversity (according to Simpson's index) increased with synchrony (Fig. 2i), and diversity (according to Simpson's index) was associated with lower temporal stability of producers, consumers, and predators (Extended Data Fig. 2c,2f,2i). If the authors used $1-D$, then my remarks can be ignored. (Since the authors found that their version was highly correlated with Shannon's index, I am guessing that they used $1-D$).

Lines 182-183. "The biological mechanism increasing the diagonal entries for predators could involve effects on density-dependent (e.g., consumption rates) or density-independent contributions to the per capita growth rate of predators (e.g., intrinsic growth rates, which are generally negative values for predators)." These are reasonable points, but the authors should have followed up with tracking down some supportive data for that sort of interpretation. For example, in a model of phytoplankton-zooplankton interaction, Norberg and DeAngelis (1997) used available data on Q10 values for zooplankton grazing rate ($Q_{10} = 3.1$, Kersting 1978), respiration ($Q_{10} = 2.59$, Goss and Bunting 1980), and losses ($Q_{10} = 2.59$, Wulff 1980). The fact that grazing rate has a higher Q10 than the respiration and losses would seem to support that increased temperature would increase the trace of the Jacobian. In the model of Norberg and DeAngelis (1997), an increase in temperature could lead to a Hopf bifurcation.

Lines 206-213. The authors' results indicate that warmer temperatures lead to higher species synchrony. They cite some previous empirical studies that agree with those results, but also cite studies that do not agree. So, there do not appear to be strong ecological reasons for the results of the current study. The authors should try to explore this further, as these results on synchrony are interesting and it would be good if these could be explained and shown to be consistent with previous results.

This is new and interesting work. The EDM analysis of the time series seems correct, though the procedures are complex, and I could not follow everything. The problem (for me) is in the interpretation of results. The results on temperature dependence of stability make sense and seem to agree with some analytic modeling. Although there are potential confounding factors in natural

systems, especially differences in nutrient levels, which have a strong effect on stability, the effects of temperature seem fairly strong. However, the effects of biodiversity are not entirely clear to me at the moment. The authors need to give more information on the Simpson's index that they used. If they used 1-D, then the effect of the two biodiversity estimates, species richness and Simpson's index, are relatively consistent and there are no problems with the biodiversity estimates that I can see. If they used D, then there would be a problem; that is species richness and Simpson's diversity have opposite effects on stability and synchrony. I think the authors should do a lot more to interpret their results from the viewpoint of data and modeling that exist, some of which I mentioned above.

Minor comments

Line 49. 'structural stability' should be defined here

Lines 69-72. This sentence is awkward.

Line 86. Change 'real-word' to 'real-world'

Line 141. Change 'structural of food webs' to 'structural stability of food webs'

Line 157. Define 'synchrony' in the context used here.

Lines 202-205. "However, effects of Simpson diversity on structural stability are not trivial because species abundance distributions and the trace of the interaction matrix are both the consequence of the underlying population dynamics." This sentence seems like a non sequitur. Is there a missing sentence before this?

Line 452. I think that 'had their greatest proportion' should be 'had a greater proportion'

Line 545. Change 'webs' to 'web'

Line 616, and elsewhere. Change 'lake Mendota and Trout' to 'Lake Mendota and Trout Lake'

Reviewer #2 (Remarks to the Author):

Please see attached document.

Reviewer #3 (Remarks to the Author):

I have reviewed the paper "Long-term effects of temperature and biodiversity on stability of natural aquatic food webs." This paper compiles an impressive dataset from 19 long-term studies of aquatic food webs to investigate the relationships between temperature, biodiversity and stability. The dataset used is impressive, as well as the analysis involved in constructing the food webs in order to determine structural stability on top of the more common measure of temporal stability (CV). The authors found varying relationships between temperature, different diversity metrics, and both measures of stability, which are described well throughout the manuscript.

Overall I feel like this paper is a useful contribution to the literature, and that these types of large-scale analyses are important for developing our understanding of the impacts of global change. Without a doubt, compiling this dataset and the analysis associated with constructing the food webs was a considerable amount of work. I praise the authors in this, and don't have any comments on the technical aspects of the analysis. However, I do feel like this paper left me with more questions remaining than knowledge/insight gained. Often this is a sign of a good paper, and I do think that in large part this is the case here, however I do think that addressing some of the questions I raise below would provide much more insight into mechanisms behind the stability of

food webs and make this a much stronger contribution to the literature.

As it currently stands, this paper reads almost as a listing of correlative results and there is little discussion of mechanisms driving these, aside from moderate references to previous di/similar results from the literature. Further investigation into these mechanisms, along with deeper embedding of the theoretical context or nuances associated with the results found here, would much better contribute to the conceptual understanding of **how** food webs function.

That said, I do think that this is certainly attainable given the data available and the backbone of work the authors have already accomplished (i.e., the food web constructions). Below, I list specific comments on the manuscript, most of which are aligned with this general impression. As such, many of my comments below are phrased as questions that were raised while reviewing this paper, and I would encourage the authors to consider addressing (or at least acknowledging in the discussion) these.

Specific comments:

Line 60: define "contraction rate", in relatable terms if possible, here. Given this is a major point of this manuscript, and that it is being introduced so early in the ms as a measure of stability, it would be helpful for a general audience to know specifically (or even qualitatively) what this measurement is referring to.

Introduction: I have no issues with the information presented in the introduction, but I am left wanting more context in places and more nuance. For example, there are clearly nuanced and context-dependent reasons why temperature may have differential effects on different populations or ecosystem types, and therefore various measures of stability, but it is presented as if the relationship is just noisy or unknown with little mention of potential mechanisms behind the variable effects. Related to this, biodiversity loss could very well be driven by temperature-induced effects on stability (the ultimate loss in stability: species extinction/extirpation), so it is not clear to me – based on the information and context presented in the introduction – why these drivers are being separated. Ultimately, providing more context and addressing nuances like these will make the motivation behind this (clearly important) study to be more clear.

Line 141: add "stability" (I think) after structural.

I am surprised there was no relationship between latitude and structural stability, given the relationship with temperature. I realize season was treated as a random effect in the modeling, but I am curious if there is an aspect to seasonality (e.g., **strength** of seasonality) that could help to explain some of this?

I am curious why a 1.5 year window was chosen (e.g., over an even number) based on the likely influence of seasonality on temporal variation?

I appreciate you stating that length of study period does not impact the general results, but I am curious if temperature or biodiversity change significantly over the monitoring period and if this matters?

Can you speak to whether the changes in temporal stability (CV) are mean- or variance-driven? This especially comes to mind concerning differences in CV across latitude.

I like the addition of an investigation into synchrony – I think this is an important factor that may help explain differences in temporal stability. But as presented in the results (given the format of this journal, a reader may not read the methods before the results), it seems to somewhat come out of nowhere and I would appreciate an earlier mention of this analysis, including what scale synchrony is being measured at.

Related to this measure of synchrony, since you see differential impacts of trophic classes on stability, why not investigate differential effects on synchrony in these food webs (e.g., predators vs consumers vs prey) species, to see how different groups are contributing to overall community

synchrony? This seems like a clear extension and would greatly aid in the interpretation of the results and insight into what is driving the large scale patterns found in this paper.

Line 209: consumption "by" predators ?

Line 223: The fact that this analysis is restricted to planktonic species is a *very* important point, for various reasons, some of which you discuss in this paragraph. Though I agree that many general conclusions can be made through interpretation of this study's results, I think that this is still quite an important piece of information that should be included much earlier in the paper.

Line 419: Do you mean 18 datasets?

Line 536: How do these numbers compare to species richness (or, more appropriately, s^2 as a more meaningful measure of connectance in the food webs)?

The above comment relates to the general feeling that I get, which is that more insight could be (quite easily, at least it appears so) gained from a little more digging. For example, do certain aspects of food web structure (e.g., connectance, food chain length, trophic biomass structure), which are already available given the dataset and current results, help to describe why certain relationships occur, or add information as to why certain relationships/measurements are variable or not consistent? These aspects would get much closer to explaining why the observed patterns are being seen and have important, and general, implications for food web functioning in other systems.

Overall: When reading this paper, I found myself scrolling around a lot to find information (between results, methods and supplement; and often back to the results to check for something I may have missed). There is a lot of information here, and I wonder if a schematic figure or table would help summarize the general relationships (or lack of, in the case of some variables) found here.

Review

Zhao et al (2022) *Nature Communications*

Comments to Authors

Zhao and collaborators investigate the question of how temperature, species richness, and a diversity index may be associated with two stability properties (structural and temporal stability) in natural food webs. I think that the question is interesting and that the authors approach it by combining an impressive data set with state-of-the-art population time series methods. Their main result is that temperature is associated with a higher structural and temporal stability across food webs (Figure 1a and 1d). However, species richness and the diversity index are not consistently associated with both stability properties (Figure 1b, 1c, 1e, and 1f). The authors also provide some explanations for these results in terms of contributions of different trophic levels (structural stability; Figure 2a-f) and of synchrony of species abundances (temporal stability; Figure 2g-i). As described below, I have three major criticisms of this work as well as multiple minor comments. I look forward to receiving a revised version of this manuscript that addresses these important issues.

Major points

1. My main criticism is related to potential problems with comparing $\text{Tr}(\mathbf{J})$ (VCR) across food webs. As the authors explain, the Jacobian coefficients are estimated using the MDR S-map, which relies on 3 hyperparameters (θ , λ , and α) and these hyperparameters certainly affect $\text{Tr}(\mathbf{J})$. For example, $\text{Tr}(\mathbf{J})$ can be lower simply because of a higher regularization parameter that shrinks large regression coefficients. From what I understood, these hyperparameters are estimated separately for each food web. Therefore, the relationship shown in Figure 1a could be an artifact of differences in how $\text{Tr}(\mathbf{J})$ is estimated across food webs. I do not think that including sampling location as a random effect solves this problem as it is related to computing the response variable in a different way for each level of the random effect. To me, it seems much more reasonable to evaluate the effect of temperature on $\text{Tr}(\mathbf{J})$ for each food web separately, where we know that θ , λ , and α are fixed. Thus, I wonder if the authors performed this analysis for each food web separately and if the effect of temperature on $\text{Tr}(\mathbf{J})$ remains the same for each food web. I suggest the authors to include this analysis to check whether the effect of temperature on $\text{Tr}(\mathbf{J})$ is consistent across individual food webs.
2. I apologize if I missed this, but it is not clear to me if and how species richness is changing through time for a given food web and the consequences of this for the analyses. For example, temperature, $\text{Tr}(\mathbf{J})$, and CV are all changing through time and the authors can analyze relationships between them over time for any given food web (as I suggested above). However, is richness also changing over time or is it only changing across different food webs? If it is changing over time for a given food web,

I am puzzled with how the authors perform the EDM analyses, since they would not have time series for all species for all time. If it is not changing over time for any given food web, then this analysis only makes sense via the cross-system comparison that the authors use. That is, the effect of richness on $\text{Tr}(\mathbf{J})$ cannot be explored for an individual food web with a constant richness over time (the approach I suggest above). I believe this is an important issue for the authors to clarify.

3. I would suggest the authors to include a few sentences in the Discussion about the differences between structural ($\text{Tr}(\mathbf{J})$) and temporal (CV) stability and about how temperature may affect these two measures of stability, given their differences. For example, $\text{Tr}(\mathbf{J})$ measures the local sensitivity of abundance trajectories (i.e., volume contraction in state space) to small perturbations. It is a property of how a dynamical system would respond to perturbations. On the other hand, CV is a statistical measure of the magnitude of fluctuations with respect to the mean. Thus, CV may be high or low due to a number of factors including intrinsic dynamics, noise, or perturbations. I think these differences have to be clear in the Discussion as well as an explanation of what does it mean for temperature to increase both measures.
4. This is more of a suggestion than a criticism. I think that including a figure before Figures 1 and 2 illustrating the time-series analyses will be very helpful. For example, the authors could show one of their time series and illustrate the computation of $\text{Tr}(\mathbf{J})$ and CV over time. Then, they could also show the corresponding temperature time series as well as the food web with species richness (e.g., as in Figure S5). With this figure it would be much easier to see what is being measured for each food web to produce the plots in Figures 1 and 2.

Minor points

- Line 1: I wonder if “long-term” is the best term to use in the title. From what I understand, the authors are trying to convey the idea that they analyze long time series. However, this can also be interpreted as how temperature (or biodiversity) have long-lasting effects on stability, which is not what is studied here. Perhaps removing this word or thinking of an alternative wording would make the title more precise.
- Line 37: species abundances instead of community dynamics?
- Line 50: I am not sure if I agree with the theoretical/empirical division here for these two measures of stability. For example, Cenci & Saavedra (2019, *Nat Ecol Evol*) or Chang et al (2021, *Ecol Lett*) contain applications of the volume contraction rate to empirical data. I would highlight the differences between these two measures of stability (see major point 3 above), instead of theoretical vs empirical work.
- Line 61: Cenci & Saavedra (2019, *Nat Ecol Evol*) have found a causal link between temperature and volume contraction rate in an empirical community. Although I believe it is not a food web, it would be worthwhile to briefly mention this result here.
- Line 86: I believe “natural” would be more precise here than “real-world”

- Line 87: Daily instead of diurnal?
- Line 110: I would change “food web topology” to something more precise. If the authors are referring to CCM here, from my understanding this method can only detect whether two species belong to the same dynamical system and, therefore, cannot uncover network topology.
- Line 112: I believe “community matrix” is used for the Jacobian matrix evaluated at a fixed point. I would use “Jacobian matrix” instead.
- Line 122: As mentioned in the previous comment, I would use “Jacobian matrix” here as well. This can avoid confusions between the Jacobian matrix, which contains local aggregated effects between species, and an interaction matrix of a parametric model (e.g. Lotka-Volterra model).
- Line 141: structural stability?
- Line 147: I would just clarify that “predator dynamics” corresponds to the aggregated effect of other species on predator species. This is because the i th diagonal coefficient in the Jacobian matrix may reflect an aggregated effect of multiple species on species i . I would also clarify this in the corresponding Methods section.
- Line 157: Synchrony of what?
- Line 174: How about “contribution of species effects on predators and consumers”?
- Line 198: Because diagonal elements can be either positive or negative, their sum will not necessarily increase when richness increases.
- Line 429: Here the authors state that time series had at least 35 points, but from Table S2 it seems that all time series had a much higher number of points. Could the authors please clarify this? If this is because there are gaps in the time series, could the authors explain how they dealt with missing data in their EDM analyses?
- Line 539: I would mention somewhere in this section that the regression coefficients approximate the coefficients of the discrete-time Jacobian matrix that are given by $\frac{\partial x_i(t+1)}{\partial x_j(t)}$. I would also break the first sentence in two. Also use “Jacobian matrix” instead of “Jacobian”.
- Line 594: I would use the notation $\text{Tr}(J)$ through the text, because Tr is simply the notation for the trace.
- Line 628: log transformed?
- Figure 1: It is a bit confusing how increasing values of $\text{Tr}(J)$ and CV are associated with a lower structural stability and temporal stability, respectively. I would suggest keeping only $\text{Tr}(J)$ and CV in the y-axis (what you actually measure) and then add an arrow pointing down with something like “higher structural (temporal) stability”.

- Figure 2: Check whether y-axis names are consistent with what is used in the text. I believe you used “contribution from predators” in the text.

REVIEWER COMMENTS

Responses to Reviewer #1

(The Line numbers below refer to the color-highlighting version of manuscript)

Reviewer #1 (Remarks to the Author):

The authors apply empirical dynamic modeling (EDM) to determine effects of temperature and biodiversity on the structural stability and temporal stability of natural aquatic systems. They take advantage of the fact that there are long-term time series from 19 aquatic food webs, including lakes, rivers and coastal marine sites. The authors use convergent cross mapping to infer species interactions and S-maps to estimate interaction strengths through time. This allows both temporal and structural stability to be estimated from the constructed Jacobian, where recent theory, volume contraction rate, is used to estimate the structural stability from the trace of the Jacobian.

Comments

Lines 86-90. “First, temperature dynamics in nature include diurnal, seasonal, and annual fluctuations and span broad temperature gradients, which can influence ecological stability in a diversity of ways that may not be feasibly studied in experimental approaches.” I don’t agree here. Mesocosm experiments can be performed in which temperature can be varied in almost any desired way to simulate seasonal variation, etc. In fact, there is an advantage in such experiments, which can exclude many possible confounding factors, such as nutrient concentrations.

Re1: Thank you for pointing this out. In the revised manuscript, we have deleted the whole sentence.

Lines 136-137. “Simpson diversity was only associated with higher structural and temporal stability (Figure 1c,f).” Lines 158-159. ‘Simpson diversity decreased with synchrony (Fig. 2h-i)’ Line 166 ‘but Simpson diversity was associated with higher temporal stability of producers, consumers, and predators (Extended Data Fig. 2c, 2f, 2i)’. The authors need to state exactly which version of Simpson’s diversity index is used. As far as I can see, the authors do not show the formula they used. The basic form of Simpson’s diversity index is D . Diversity decreases with increases in Simpson’s index, D . Diversity is zero when Simpson’s index is 1. If that is what the authors used, then, my reading from the figures is that temporal stability decreased with diversity (according to Simpson’s index) (Fig. 1f), diversity (according to Simpson’s index) increased with synchrony (Fig. 2i), and diversity (according to Simpson’s index) was associated with lower temporal stability of producers, consumers, and predators (Extended Data Fig. 2c,2f,2i). If the authors used $1-D$, then my remarks can be ignored. (Since the authors found that their version was highly correlated with Shannon’s index, I am guessing that they used $1-D$).

Re2: We apologize that we didn’t state the formula of Simpson diversity in the previous manuscript. In the previous version, we indeed computed the Simpson index as $1-D$. In the revised manuscript, we explicitly show the formula (lines 662-664).

Lines 182-183. “The biological mechanism increasing the diagonal entries for predators could involve effects on density-dependent (e.g., consumption rates) or density-independent contributions to the per capita growth rate of predators (e.g., intrinsic growth rates, which are generally negative values for predators).” These are reasonable points, but the authors should have followed up with tracking down some supportive data for that sort of interpretation. For example, in a model of phytoplankton-zooplankton interaction, Norberg and DeAngelis (1997) used available data on Q10 values for zooplankton grazing rate (Q10 = 3.1, Kersting 1978), respiration (Q10 = 2.59, Goss and Bunting 1980), and losses (Q10 = 2.59, Wulff 1980). The fact that grazing rate has a higher Q10 than the respiration and losses would seem to support that increased temperature would increase the trace of the Jacobian. In the model of Norberg and DeAngelis (1997), an increase in temperature could lead to a Hopf bifurcation.

Re3: In the revised manuscript, we have added the work of Norberg and DeAngelis (1997) as a support (lines 221, 225). In addition, we have also added other references that support this section (lines 209-223).

Lines 206-213. The authors’ results indicate that warmer temperatures lead to higher species synchrony. They cite some previous empirical studies that agree with those results, but also cite studies that do not agree. So, there do not appear to be strong ecological reasons for the results of the current study. The authors should try to explore this further, as their results on synchrony are interesting and it would be good if these could be explained and shown to be consistent with previous results.

Re4: In the revised manuscript, we have explored this result further by doing an additional literature study (lines 243-252).

This is new and interesting work. The EDM analysis of the time series seems correct, though the procedures are complex, and I could not follow everything. The problem (for me) is in the interpretation of results. The results on temperature dependence of stability make sense and seem to agree with some analytic modeling. Although there are potential confounding factors in natural systems, especially differences in nutrient levels, which have a strong effect on stability, the effects of temperature seem fairly strong. However, the effects of biodiversity are not entirely clear to me at the moment. The authors need to give more information on the Simpson’s index that they used. If they used 1-D, then the effect of the two biodiversity estimates, species richness and Simpson’s index, are relatively consistent and there are no problems with the biodiversity estimates that I can see. If they used D, then there would be a problem; that is species richness and Simpson’s diversity have opposite effects on stability and synchrony. I think the authors should do a lot more to interpret their results from the viewpoint of data and modeling that exist, some of which I mentioned above.

Re5: The Simpson’s diversity we used is 1-D, (cf. **Reply2**). In the revised manuscript, we have worked on better interpreting our results on the temperature effects (lines 169-174, 181-184, 243-245), and added potential explanations of the biodiversity effects (species richness and Simpson’s diversity) on stability (lines 169-171, 172-174, 181-184, 245-252).

Minor comments

Line 49. ‘structural stability’ should be defined here

Re6: As suggested, we have defined it in the revised manuscript (lines 54-61).

Lines 69-72. This sentence is awkward.

Re7: We have rewritten this sentence in the revised manuscript.

Line 86. Change ‘real-word’ to ‘real-world’

Re8: Thanks for pointing out the spelling error. In revised manuscript, we have changed “real-word” to “natural”. The term “natural” here looks more appropriate than “real-world”. (Please also see **Reply24**)

from Reviewer#2)

Line 141. Change 'structural of food webs' to 'structural stability of food webs'

Re9: We have changed it accordingly.

Line 157. Define 'synchrony' in the context used here.

Re10: In the revised manuscript, we have defined the 'synchrony' as *synchrony of all species within the food web* (line 170).

Lines 202-205. "However, effects of Simpson diversity on structural stability are not trivial because species abundance distributions and the trace of the interaction matrix are both the consequence of the underlying population dynamics." This sentence seems like a non sequitur. Is there a missing sentence before this?

Re11: In the revised manuscript, we have reorganized this sentence and re-illustrated the effects of Simpson diversity (lines 235-237).

Line 452. I think that 'had their greatest proportion' should be 'had a greater proportion'

Re12: We have changed it accordingly (line 491).

Line 545. Change 'webs' to 'web'

Re13: We have changed it accordingly (line 591).

Line 616, and elsewhere. Change 'lake Mendota and Trout' to 'Lake Mendota and Trout Lake'

Re14: We have changed it accordingly throughout manuscript.

Responses to Reviewer #2

(The Line numbers below refer to the color-highlighting version of manuscript)

Reviewer #2 (Remarks to the Author):

Please see attached document.

Comments to Authors

Zhao and collaborators investigate the question of how temperature, species richness, and a diversity index may be associated with two stability properties (structural and temporal stability) in natural food webs. I think that the question is interesting and that the authors approach it by combining an impressive data set with state-of-the-art population time series methods. Their main result is that temperature is associated with a higher structural and temporal stability across food webs (Figure 1a and 1d). However, species richness and the diversity index are not consistently associated with both stability properties (Figure 1b, 1c, 1e, and 1f). The authors also provide some explanations for these results in terms of contributions of different trophic levels (structural stability; Figure 2a-f) and of synchrony of species abundances (temporal stability; Figure 2g-i). As described below, I have three major criticisms of this work as well as multiple minor comments. I look forward to receiving a revised version of this manuscript that addresses these important issues.

Major points

1. My main criticism is related to potential problems with comparing $\text{Tr}(J)$ (VCR) across food webs. As the authors explain, the Jacobian coefficients are estimated using the MDR S-map, which relies on 3 hyperparameters (θ , λ , and α) and these hyperparameters certainly affect $\text{Tr}(J)$. For example, $\text{Tr}(J)$ can be lower simply because of a higher regularization parameter that shrinks large regression coefficients. From what I understood, these hyperparameters are estimated separately for each food web. Therefore, the relationship shown in Figure 1a could be an artifact of differences in how $\text{Tr}(J)$ is

estimated across food webs. I do not think that including sampling location as a random effect solves this problem as it is related to computing the response variable in a different way for each level of the random effect. To me, it seems much more reasonable to evaluate the effect of temperature on $\text{Tr}(J)$ for each food web separately, where we know that θ , λ , and α are fixed. Thus, I wonder if the authors performed this analysis for each food web separately and if the effect of temperature on $\text{Tr}(J)$ remains the same for each food web. I suggest the authors to include this analysis to check whether the effect of temperature on $\text{Tr}(J)$ is consistent across individual food webs.

Re15: Thank you very much for this constructive remark and valuable suggestion.

As you pointed out, the 3 hyperparameters (θ , λ , and α) were indeed estimated separately for each food web. Since the 3 hyperparameters were different across food webs, there is indeed a risk that temperature effects on the 3 hyperparameters would cause the relationship between temperature and structural stability $\text{Tr}(J)$. We have therefore tested, in our revised paper, whether this is happening. Specifically, we tested 1) the effect of temperature on each of the 3 hyperparameters; and 2) the effect of each of the 3 hyperparameters on $\text{Tr}(J)$. Our analyses show that 1) there is no relationship between temperature and the 3 hyperparameters (Fig. R1); 2) there is no relationship between the 3 hyperparameters and $\text{Tr}(J)$ (Fig. R2) in this study.

As you requested, we also examined the effect of temperature on $\text{Tr}(J)$ for each food web separately. We found that the effect of temperature on $\text{Tr}(J)$ is system dependent (Fig. 2a). Yet, in 13 out of 19 food webs, temperature had positive effects on $\text{Tr}(J)$. In the remaining 6 food webs, temperature had negative effects.

Overall, we concluded: 1) the 3 hyperparameters (θ , λ , and α) in the MDR S-map did not cause the temperature effect on stability in this study; 2) within the range considered, temperature is positively related to $\text{Tr}(J)$ across 19 food webs, but within each food web, the effect appears to be system dependent.

Figure R1. The effect of geometric mean temperature (a-c) and variance of temperature (d-e) on each of the three parameters (α , θ , or λ) across 19 food webs. The effects are examined by linear mixed models, which is the same statistical model as in the main text. The three parameters (α , θ , and λ) are used to estimate the Jacobian matrix, during MDR S-map analysis. Unlike temperature, which changes over time, here α , θ and λ remain constant over time. Thus, in linear mixed models, we compute the geometric mean of temperature and the variance of temperature as fixed factors, treat each of the three parameters (α , θ , or λ) as the response variable, and treat sampling location (e.g. Lake Mendota and Trout Lake) as a random effect. During MDR S-map analysis to estimate Jacobian elements, each network node within a food web will generate a single value for each of three parameters (α , θ , or λ). Thus, within this food web, the number of nodes used to MDR S-map analysis is the same as the number of each of the three parameters (α , θ , or λ), e.g. the number of α is 35 in Lake Monona in Fig 1R.a, which is same as the number of network nodes used to MDR S-map analysis, as shown in Fig S9. Dashed lines indicate nonsignificant effects.

Figure R2. The effect of geometric mean of α (a), geometric mean of θ (b) geometric mean of λ (c), variance of α (d), variance of θ (e), and variance of λ (f) on structural stability $Tr(J)$ across 19 food webs. The effects are examined by linear mixed models, which is the same statistical model as in the main text. Unlike the structural stability $Tr(J)$, which changes over time, here α , θ and λ remain constant over time. Thus, in the linear mixed models, we compute the geometric mean of α (a), the variance of α (d), the geometric mean of θ (b), the variance of θ (e), the geometric mean of λ (c), and the variance of λ (f) as fixed factors, and treat $Tr(J)$ as a response variable, and treat sampling location (e.g. Lake Mendota and Trout Lake) as a random effect. Dashed lines indicate nonsignificant effects.

Figure 2a. The effect of temperature on structural stability $Tr(J)$. A smaller $Tr(J)$ indicates higher stability. Coloured points correspond to values at each food web in each season of each year. The black bold line indicates a significant effect of temperature across 19 food webs, while non-bold coloured lines indicate effects of temperature within a food web. Shaded areas represent 95% confidence intervals across 19 food webs. For statistical results see table S1.

2. I apologize if I missed this, but it is not clear to me if and how species richness is changing through time for a given food web and the consequences of this for the analyses. For example, temperature, $Tr(J)$, and CV are all changing through time and the authors can analyze relationships between them over time for any given food web (as I suggested above). However, is richness also changing over time or is it only changing across different food webs? If it is changing over time for a given food web, I am puzzled with how the authors perform the EDM analyses, since they would not have time series for all species for all time. If it is not changing over time for any given food web, then this analysis only makes sense via the cross-system comparison that the authors use. That is, the effect of richness on $Tr(J)$ cannot be explored for an individual food web with a constant richness over time (the approach I suggest above). I believe this is an important issue for the authors to clarify.

Re17: Thank you for this key comment. We apologize that this was not made clear in previous version. In our revised manuscript, we now clearly state that for each food web, we are tracking how species richness (i.e. the sampled species richness observed at any point in time) is changing over time, and explain how we perform the EDM analyses. Because species abundance can have both zero and nonzero positive values (presence and absence) over time, species richness (i.e. the sampled species richness) is useful as it reflects changes in the underlying relative abundances of species (historically referred to as *community structure* (Pielou 1966, Sugihara 1980, Sugihara et al 2003)).

We make a distinction between the *sampled species richness*, and *number of network nodes S* in a food web which is independent of time and used to perform EDM analysis.

In our manuscript, the *sampled species richness* is used to test the relationship with $\text{Tr}(J)$ and CV, as explained in the main text (lines 664-666). The *number of network nodes* S in a food web (i.e. fixed value of S) is the size of Jacobian matrix ($S \times S$, row \times column), and therefore the size S is fixed over time, as shown in the MDR S-map technique (Chang et al. 2021). We show this in the new schematic Figure 1. For the Jacobian matrix, there is an associated adjacency matrix A (i.e. the Zero-One matrix $N \times N$, showing the binary causal network estimated by convergent cross mapping CCM). A value of 1 indicates a valid causal link, but at each point in time the species interaction strength of the link (i.e. corresponding elements in Jacobian matrix) is estimated by MDR S-map technique. In contrast, the value of 0 indicates no link, and therefore the corresponding element in Jacobian matrix is estimated as 0 in the MDR S-map (Chang et al. 2021).

As an example, take Oneida Lake, which is one of the lakes we used in our manuscript and was also used in recent studies (Rogers et al. 2022; Chang et al. 2022). The *number of network nodes* S is 26 (abundance of each of the 26 species shown at Fig R3, interaction network shown at Fig S9). The *sampled species richness* varies through time ranging from 4 to 26 (Fig R4). Because the zero and nonzero abundance can affect the estimation of the values of $\text{Tr}(J)$ using MDR S-map, we used the time series of the *sampled species richness* to test its effects on $\text{Tr}(J)$, as shown in Fig 2. To summarize, the size of the Jacobian matrix, which always remains constant at 26×26 over time, is determined by the adjacency matrix estimated by CCM. Then, the elements in the Jacobian matrix were estimated by MDR S-map.

Finally, in the revised manuscript, and for reasons explained in the previous reply (see **Reply15**), we tested if biodiversity (species richness and Simpson) affects the 3 hyperparameters (θ , λ , and α). We found there is no relationship between biodiversity and the 3 hyperparameters (Fig. R5). In addition, the effect of species richness on $\text{Tr}(J)$ was also system dependent albeit mostly positive, as are the temperature effects. In 14 out of 19 food webs, species richness had positive effects on $\text{Tr}(J)$, and in the remaining 5 food web, species richness had negative effects (Fig 2b).

Rogers, T.L., Johnson, B.J. & Munch, S.B. (2022) Chaos is not rare in natural ecosystems. *Nature Ecology & Evolution*, 6, 1–7.

Chang, C.W., Miki, T., Ye, H., Souissi, S., Adrian, R., Anneville, O., Agasild, H., Ban, S., Be'eri-Shlevin, Y., Chiang, Y.R. and Feuchtmayr, H., 2022. Causal networks of phytoplankton diversity and biomass are modulated by environmental context. *Nature communications*, 13,1-11.

Chang, C. W., Miki, T., Ushio, M., Ke, P. J., Lu, H. P., Shiah, F. K., & Hsieh, C. H. (2021). Reconstructing large interaction networks from empirical time series data. *Ecology Letters*, 24, 2763-2774.

Figure R3. Abundance (individuals/litre) for 26 species over time in Oneida Lake. Data is seasonal frequency (4 points per year). The horizontal black dashed line represents zero abundance.

Figure R4. Species richness and Simpson diversity over time in Oneida Lake. Data is seasonal frequency (4 points per year).

Figure R5. The effect of the geometric mean of species richness (a-c) and geometric mean of Simpson (d-e) on each of the three parameters (α , θ , or λ) across 19 food webs. The effects are examined by linear mixed models, which is the same statistical model as in the main text. The three parameters (α , θ , and λ) are used to estimate the Jacobian matrix, during MDR S-map analysis. Unlike the species richness and Simpson diversity, which all change over time, here α , θ and λ remain constant over time. Thus, in the linear mixed models, we compute the geometric mean of species richness and Simpson as fixed factor, treat each of the three parameters (α , θ , or λ) as a response variable, and treat sampling locations (e.g. Lake Mendota and Trout Lake) as random effects. During MDR S-map analysis to estimate Jacobian elements, each network node within a food web will generate a single value for each of three parameters (α , θ , or λ). Thus, within this food web, the number of nodes used to do MDR S-map analysis is the same as the number of each of three parameters (α , θ , or λ), e.g. the number of α is 35 in Lake Monona in Fig 1R.a, which is the same as the number of network node used to MDR S-map analysis, as shown in Fig S9). Dashed lines indicate nonsignificant effects.

Figure 2b-c. The effect of species richness (b) and Simpson diversity (c) on structural stability $Tr(J)$. A smaller $Tr(J)$ indicates higher stability. Coloured points correspond to values at each food web in each season of each year. The black bold line indicates significant the effect of temperature across 19 food webs. Non-bold coloured lines indicate the effects of temperature within each food web. Shaded areas represent 95% confidence intervals across 19 food webs.

3. I would suggest the authors to include a few sentences in the Discussion about the differences between structural ($Tr(J)$) and temporal (CV) stability and about how temperature may affect these two measures of stability, given their differences. For example, $Tr(J)$ measures the local sensitivity of abundance trajectories (i.e., volume contraction in state space) to small perturbations. It is a property of how a dynamical system would respond to perturbations. On the other hand, CV is a statistical measure of the magnitude of fluctuations with respect to the mean. Thus, CV may be high or low due to a number of factors including intrinsic dynamics, noise, or perturbations. I think these differences have to be clear in the Discussion as well as an explanation of what does it mean for temperature to increase both measures.

Re18: In the revised manuscript, we have pointed out the difference between structural and temporal stability, and how temperature may affect these two measures. We do so in the Introduction (lines 56-63, 65-73, 74-76) and in the Discussion (205-206, 241-242) section.

4. This is more of a suggestion than a criticism. I think that including a figure before Figures 1 and 2 illustrating the time-series analyses will be very helpful. For example, the authors could show one of their time series and illustrate the computation of $Tr(J)$ and CV over time. Then, they could also show the corresponding temperature time series as well as the food web with species richness (e.g., as in Figure S5). With this figure it would be much easier to see what is being measured for each food web to produce the plots in Figures 1 and 2.

Re19: Thank you for your suggestion, we have added a new Figure (Fig. 1) and used Lake Monona as an example, to schematically illustrate our methodology.

Figure 1. An overview of the data and methods used in this paper, illustrated with the data from Lake Monona. In **a**, species abundances over time are shown, where the horizontal black dashed line indicates a zero species abundance (species absence). In **b**, we reconstruct the interaction network using convergent cross mapping CCM. In **c**, we compute structural stability for each time point as the trace of the local Jacobian matrix $Tr(J)$. The Jacobian matrices are inferred by multiview distance regularised S-map (MDR S-map). The size of the Jacobian is fixed over time, as $s \times s$ (row \times column), where s is the number of network nodes in the food web (i.e. $s = 35$ for the food web in Lake Monona). In **d**, the temporal stability of the food web and of each trophic group are computed as the coefficient of variation of the total community abundance and each trophic group, respectively, using a time window of 1.5 years. Species asynchrony of the food web and of each trophic group is computed using a moving window. In **e**, time varying the species richness (i.e. the sampled species richness based purely on species presence and absence) and Simpson diversity are calculated from the data, and combined with time varying temperature data.

Minor points

- Line 1: I wonder if “long-term” is the best term to use in the title. From what I understand, the authors are trying to convey the idea that they analyze long time series. However, this can also be interpreted as how temperature (or biodiversity) have

long-lasting effects on stability, which is not what is studied here. Perhaps removing this word or thinking of an alternative wording would make the title more precise.

Re20: In the revised manuscript, we have removed “long-term” from the title.

- Line 37: species abundances instead of community dynamics?

Re21: In the revised manuscript, we have changed it accordingly.

- Line 50: I am not sure if I agree with the theoretical/empirical division here for these two measures of stability. For example, Cenci & Saavedra (2019, Nat Ecol Evol) or Chang et al (2021, Ecol Lett) contain applications of the volume contraction rate to empirical data. I would highlight the differences between these two measures of stability (see major point 3 above), instead of theoretical vs empirical work.

Re22: In the revised manuscript, we have removed “the theoretical/empirical division”, and switched to highlighting the difference between the two stability indices (lines 54-63).

- Line 61: Cenci & Saavedra (2019, Nat Ecol Evol) have found a causal link between temperature and volume contraction rate in an empirical community. Although I believe it is not a food web, it would be worthwhile to briefly mention this result here.

Re23: As suggested, in the revised manuscript, we have mentioned it (lines 68-69).

- Line 86: I believe “natural” would be more precise here than “real-world”

Re24: As suggested, in the revised manuscript, we have replaced it as “natural”.

- Line 87: Daily instead of diurnal?

Re25: In the revised manuscript, we have deleted the whole sentence which includes “daily”, because reviewer#1 suggested that Mesocosm experiments can simulate seasonal variation of temperature (See **Reply 1**).

The deleted sentence is “First, temperature dynamics in nature include diurnal, seasonal, and annual fluctuations and span broad temperature gradients, which can influence ecological stability in a diversity of ways that may not be feasibly studied in experimental approaches.”

- Line 110: I would change “food web topology” to something more precise. If the authors are referring to CCM here, from my understanding this method can only detect whether two species belong to the same dynamical system and, therefore, cannot uncover network topology.

Re26: As suggested, In the revised manuscript, we have replaced “food web topology” as “interaction network” (line 118).

- Line 112: I believe “community matrix” is used for the Jacobian matrix evaluated at a fixed point. I would use “Jacobian matrix” instead.

Re27: As suggested, in the revised manuscript, we have changed it accordingly (line 120).

- Line 122: As mentioned in the previous comment, I would use “Jacobian matrix” here as well. This can avoid confusions between the Jacobian matrix, which contains local aggregated effects between species, and an interaction matrix of a parametric model (e.g. Lotka-Volterra model).

Re28: As suggested, in the revised manuscript, we have changed it accordingly.

- Line 141: structural stability?

Re29: It is indeed structural stability. In the revised manuscript, we have changed it accordingly (line 154).

- Line 147: I would just clarify that “predator dynamics” corresponds to the aggregated effect of other species on predator species. This is because the i th diagonal coefficient in the Jacobian matrix may reflect an aggregated effect of multiple species on species i . I would also clarify this in the corresponding Methods section.

Re30: In the revised manuscript, we have changed it accordingly (Lines 162-163, 694-695).

- Line 157: Synchrony of what?

Re31: In the revised manuscript, we have written it as “synchrony of all species within the food web”.

- Line 174: How about “contribution of species effects on predators and consumers”?

Re32: In the revised manuscript, we changed it to “*contributions from predators and consumers*”, as it matched the Results (lines 164, 166, 167), Discussion (lines 200, 232), and Methods (lines 695, 698-700) section.

- Line 198: Because diagonal elements can be either positive or negative, their sum will not necessarily increase when richness increases.

Re33: We have removed these sentences in the revised manuscript.

- Line 429: Here the authors state that time series had at least 35 points, but from Table S2 it seems that all time series had a much higher number of points. Could the authors please clarify this? If this is because there are gaps in the time series, could the authors explain how they dealt with missing data in their EDM analyses?

Re34: All of the time series indeed had a much longer number of points than 35. To avoid confusion, in the revised manuscript, we have removed the sentence “abundance data were sequentially collected for at least 35 sampling occasions” from the main text.

In the previous manuscript, the raw abundance data from the 19 datasets had varying sampling frequencies, changing from low frequency (seasonally sampled) to higher frequency (bimonthly, monthly, biweekly). We, therefore, converged everything to seasonal averages (lines 475-479), which allowed us to compare stability across 19 systems. Thus, the length of the time series in Table S2 was equally seasonal intervals.

To avoid confusion, in the revised manuscript, we added a new column to Tables S2, showing the raw length of the time series prior to seasonal averaging.

- Line 539: I would mention somewhere in this section that the regression coefficients approximate the coefficients of the discrete-time Jacobian matrix that are given by $\partial x_i(t+1)/\partial x_j(t)$. I would also break the first sentence in two. Also use “Jacobian matrix” instead of “Jacobian”.

Re35: In the revised manuscript, we mentioned: “the regression coefficients approximate the interaction strength in the discrete-time Jacobian matrix given by $\frac{\partial x_i(t+1)}{\partial x_j(t)}$ ”, on lines 586-588. We also break the first sentence in two and used “Jacobian matrix” instead.

- Line 594: I would use the notation $\text{Tr}(J)$ through the text, because Tr is simply the notation for the trace.

Re36: In the revised manuscript, we have used the notation $\text{Tr}(J)$ throughout the text.

- Line 628: log transformed?

Re37: In the revised manuscript, we have used the word “log transformed” accordingly.

- Figure 1: It is a bit confusing how increasing values of $\text{Tr}(J)$ and CV are associated with a lower structural stability and temporal stability, respectively. I would suggest keeping only $\text{Tr}(J)$ and CV in the y-axis (what you actually measure) and then add an arrow pointing down with something like “higher structural (temporal) stability”.

Re38: We have changed it accordingly.

- Figure 2: Check whether y-axis names are consistent with what is used in the text. I believe you used “contribution from predators” in the text.

Re39: In the revised manuscript, we have corrected y-axis names as “contribution from predators”, which is same as the main text.

Responses to Reviewer #3

(The Line numbers below refer to the color-highlighting version of manuscript)

Reviewer #3 (Remarks to the Author):

I have reviewed the paper “Long-term effects of temperature and biodiversity on stability of natural aquatic food webs.” This paper compiles an impressive dataset from 19 long-term studies of aquatic food webs to investigate the relationships between temperature, biodiversity and stability. The dataset used is impressive, as well as the analysis involved in constructing the food webs in order to determine structural stability on top of the more common measure of temporal stability (CV). The authors found varying relationships between temperature, different diversity metrics, and both measures of stability, which are described well throughout the manuscript.

Overall I feel like this paper is a useful contribution to the literature, and that these types of large-scale analyses are important for developing our understanding of the impacts of global change. Without a doubt, compiling this dataset and the analysis associated with constructing the food webs was a considerable amount of work. I praise the authors in this, and don't have any comments on the technical aspects of the analysis. However, I do feel like this paper left me with more questions remaining than knowledge/insight gained. Often this is a sign of a good paper, and I do think that in large part this is the case here, however I do think that addressing some of the questions I raise below would provide much more insight into mechanisms behind the stability of food webs and make this a much stronger contribution to the literature.

As it currently stands, this paper reads almost as a listing of correlative results and there is little discussion of mechanisms driving these, aside from moderate references to previous di/similar results from the literature. Further investigation into these mechanisms, along with deeper embedding of the theoretical context or nuances associated with the results found here, would much better contribute to the conceptual understanding of *how* food webs function.

Re40: Thank you very much for your constructive feedback. Following your suggestion, in the revised manuscript, we have provided a more mechanistic perspective and discussion (lines 213-223, 230-237, 243-252).

That said, I do think that this is certainly attainable given the data available and the backbone of work the authors have already accomplished (i.e., the food web constructions). Below, I list specific comments on the manuscript, most of which are aligned with this general impression. As such, many of my comments below are phrased as questions that were raised while reviewing this paper, and I would encourage the authors to consider addressing (or at least acknowledging in the discussion) these.

Re41: Thank you very much for your comments. In the revised manuscript, we carefully addressed your comments by adding the following analyses: 1) effect of food web structure (e.g. link density, connectance, food chain length) on two stability indices; 2) effects of species asynchrony of each trophic group (producers, consumers, or predators) on temporal stability of each trophic group; 3) effects of the species synchrony (and temporal stability) of each trophic group on the synchrony (and temporal stability) of the whole food web.

Specific comments:

Line 60: define “contraction rate”, in relatable terms if possible, here. Given this is a major point of this manuscript, and that it is being introduced so early in the ms as a measure of stability, it would be helpful for a general audience to know specifically (or even qualitatively) what this measurement is referring to.

Re42: In the revised manuscript, we have explained the “contraction rate” in lay terms upon first mention (lines 56-61).

Introduction: I have no issues with the information presented in the introduction, but I am left wanting

more context in places and more nuance. For example, there are clearly nuanced and context-dependent reasons why temperature may have differential effects on different populations or ecosystem types, and therefore various measures of stability, but it is presented as if the relationship is just noisy or unknown with little mention of potential mechanisms behind the variable effects. Related to this, biodiversity loss could very well be driven by temperature-induced effects on stability (the ultimate loss in stability: species extinction/extirpation), so it is not clear to me – based on the information and context presented in the introduction – why these drivers are being separated. Ultimately, providing more context and addressing nuances like these will make the motivation behind this (clearly important) study to be more clear.

Re43: We agree that more context should be added in the Introduction section. Thus, in the revised manuscript, we have re-written the Introduction section to state, mainly including: 1) how temperature alters ecological parameters, therefore changes the population dynamics of different trophic levels, finally alters two measures of stability (lines 65-71, 74-76); 2) the changes in synchrony of all species within the food web and the temporal stability of specific trophic levels (e.g. producers, consumers, and predators) could alter temporal stability of whole food web (lines 79-82).

We also agree that biodiversity loss could very well be driven by temperature-induced effects on stability. Surprisingly, we didn't find pervasive biodiversity loss across our 19 datasets (See **Reply47**). Therefore, it seems more appropriate to separate the biodiversity and temperature effects in the Introduction section. We chose to focus on the effects of biodiversity dynamics rather than the effects of biodiversity loss, as indicated in the Introduction section (lines 90, 88).

Line 141: add “stability” (I think) after structural.

Re44: We have changed it accordingly.

I am surprised there was no relationship between latitude and structural stability, given the relationship with temperature. I realize season was treated as a random effect in the modeling, but I am curious if there is an aspect to seasonality (e.g., *strength* of seasonality) that could help to explain some of this?

Re45: Thanks for pointing this out. We tested the effects of seasonality (i.e. strength of seasonality) on structural stability and did not find a correlation (Fig. R6).

Figure R6. The effects of the strength of seasonality on structural stability $Tr(J)$ of food webs. The effects are examined by linear mixed models, which is the same statistical model as in the main text. In linear mixed models, structural stability $Tr(J)$ was treated as a response variable, the strength of seasonality was treated as a fixed factor, and sampling location (e.g. lake Mendota and Trout Lake) was treated as a random effect. The strength of seasonality was computed as $1 - Var(R_t)/Var(R_t + S_t)$, where R_t is the remainder component and S_t is the seasonal component in Seasonal-Trend decomposition using LOESS (STL) (Hyndman & Athanasopoulos, 2018). Values close to 1 indicate a highly seasonal time series, while values close to 0 indicate a time series with little seasonality. The strength of seasonality was achieved by using the *features* function in the *feasts* package.

Hyndman, R. J., & Athanasopoulos, G. (2018). *Forecasting: principles and practice*. OTexts. Chapter 6.7, Page 266-267.

I am curious why a 1.5 year window was chosen (e.g., over an even number) based on the likely influence of seasonality on temporal variation?

Re46: In the previous manuscript, we followed Ushio et al (2018) in choosing the same length of time points (6 time point, equivalent to 1.5 years). In the previous manuscript, we also tried a time window of 3 years, which did not qualitatively change the results. As you suggested, we now also tried an even time window (2 years), and again find qualitatively similar results (Fig. R7).

Ushio M. *et al.* Fluctuating interaction network and time-varying stability of a natural fish community. *Nature*. 2018. 554, 360-3633.

Figure. R7. The effect of temperature, species richness and Simpson diversity on the temporal stability of food webs (a-c). Temporal stability was computed as the natural log-transformed coefficient of variation of the whole community abundance. A lower CV indicates higher stability. Colored points correspond to values at each food web in each moving window (window width = 2 years). The bold lines indicate significant effects of temperature across 19 food webs, while non-bold colored lines indicate the effects within each food web. Shaded areas represent 95% confidence intervals across the 19 food webs.

I appreciate you stating that length of study period does not impact the general results, but I am curious if temperature or biodiversity change significantly over the monitoring period and if this matters? Can you speak to whether the changes in temporal stability (CV) are mean- or variance-driven? This especially comes to mind concerning differences in CV across latitude.

Re47: We apologize that we didn't test if temperature or biodiversity change significantly over the monitoring period as indeed such trends can matter. We have now tested for such effects in our datasets. Overall, we didn't find pervasive patterns for temperature or biodiversity over time (Fig. R8-R10). Specifically, temperature did not significantly change in any of the 19 datasets (Fig. R8). Similarly, Simpson's diversity in 18 out of the 19 datasets did not significant change over time; it decreased over time in a single dataset only (Fig. R9). Species richness in 3 out of 19 datasets significantly increased over time; in 3 out of 19 it significantly decreased over time; but in the other data sets it did not change through time (Figure R10).

In addition, we further tested whether the changes in temporal stability (CV) are mean- or variance-driven, as you suggested. We found that temperature and biodiversity changed both the mean and the variance. Specially, temperature increased CV by increasing the variance more than it increased the mean (Fig. R11a, d). Biodiversity (species richness and Simpson) decreased CV by decreasing the variance and increasing the mean (Fig. R11b-c, e-f).

Figure. R8. The change of temperature over time in each of 19 datasets. We employed a linear regression to test the effects of time on temperature, in each of 19 datasets. The solid lines indicate significant effects, and dashed lines indicate nonsignificant effects.

Figure. R9. The change of Simpson's diversity over time in each of 19 datasets. We employed a linear regression to test the effects of time on temperature, in each of 19 datasets. The solid lines indicate significant effects, and dashed lines indicate nonsignificant effects.

Figure. R10. The change of species richness over time in each of 19 datasets. We employed a linear regression to test the effects of time on temperature, in each of 19 datasets. The solid lines indicate significant effects, and dashed lines indicate nonsignificant effects.

Figure R11. The effect of temperature, species richness and Simpson diversity on standard deviation S.D (a-c) and the mean of whole community abundance (d-f). In a-f, colored points correspond to values at each food web in each moving window (window width = 1.5 years, which is same length as the main text in Fig 2d-f). The solid lines indicate significant effects, and the dashed lines indicate nonsignificant effects. Shaded areas represent 95% confidence intervals.

I like the addition of an investigation into synchrony – I think this is an important factor that may help explain differences in temporal stability. But as presented in the results (given the format of this journal, a reader may not read the methods before the results), it seems to somewhat come out of nowhere and I would appreciate an earlier mention of this analysis, including what scale synchrony is being measured at.

Re48: In the revised manuscript, we mention synchrony of all species within the food web early on in the Introduction (lines 79, 129). Apart from the synchrony of all species within the food web, we also introduce the temporal stability of specific trophic groups (producers, consumers or predators) (lines 80-82, 127-129). It is expected that temperature can also affect temporal stability of the whole food web by disproportionately affecting temporal stability of specific trophic groups. We hope this helps readers understand our results better.

Related to this measure of synchrony, since you see differential impacts of trophic classes on stability, why not investigate differential effects on synchrony in these food webs (e.g., predators vs consumers vs prey) species, to see how different groups are contributing to overall community synchrony? This seems like a clear extension and would greatly aid in the interpretation of the results and insight into what is driving the large scale patterns found in this paper.

Re49: In the revised manuscript, we have tested the effects of temperature and biodiversity on the synchrony of each trophic group (i.e. predators, consumers or consumers) (Fig S6a-i), and we found that species synchrony within producers determined the synchrony of all species within the food web more than did synchrony within consumers or predators (Fig S5d-f). We stated these findings in the Results section (lines 175-180) and in the discussion section (lines 249-252).

Line 209: consumption “by” predators ?

Re50: We have changed it accordingly.

Line 223: The fact that this analysis is restricted to planktonic species is a *very* important point, for various reasons, some of which you discuss in this paragraph. Though I agree that many general conclusions can be made through interpretation of this study’s results, I think that this is still quite an important piece of information that should be included much earlier in the paper.

Re51: In the revised manuscript, we now mention this in the Abstract (line 34), the Introduction (line 116), and the Discussion (line 205).

Line 419: Do you mean 18 datasets?

Re52: Yes. We have changed it accordingly.

Line 536: How do these numbers compare to species richness (or, more appropriately, s^2 as a more meaningful measure of connectance in the food webs)?

Re53: In the revised manuscript, we quantify the changes in connectance (lines 580-582).

The above comment relates to the general feeling that I get, which is that more insight could be (quite easily, at least it appears so) gained from a little more digging. For example, do certain aspects of food web structure (e.g., connectance, food chain length, trophic biomass structure), which are already available given the dataset and current results, help to describe why certain relationships occur, or add information as to why certain relationships/measurements are variable or not consistent? These aspects would get much closer to explaining why the observed patterns are being seen and have important, and general, implications for food web functioning in other systems.

Re54: As you suggested, we carried out additional analyses on food web structure (i.e. links per species, connectance and food chain length). We found: 1) link density increased the temporal stability of food webs (Fig S7d), while the other structural aspects (i.e. connectance and food chain length) had no effects on either temporal or structural food web stability (Fig S8a-d); 2) higher mean temperature was associated with a lower link density, which reduced temporal stability of food webs (Fig S7a, d). In contrast, higher mean species richness was related to higher link density, which then increased temporal stability of food webs (Fig S7b, d). We added these finding to the results section (lines 187-195) and discussion section (line 244).

Overall: When reading this paper, I found myself scrolling around a lot to find information (between results, methods and supplement; and often back to the results to check for something I may have missed). There is a lot of information here, and I wonder if a schematic figure or table would help summarize the general relationships (or lack of, in the case of some variables) found here.

Re55: We agree, and this comment echoes the one made by Reviewer #2. We have added a schematic figure (Figure 1) in the main text to illustrate the analyses we carried out (See also **Reply19**). We hope this helps getting an overview of the various methods used in the manuscript.

Figure 1. An overview of the data and methods used in this paper, illustrated with the data from Lake Monona. In **a**, species abundances over time are shown, where the horizontal black dashed line indicates a zero species abundance (species absence). In **b**, we reconstruct the interaction network using convergent cross mapping CCM. In **c**, we compute structural stability for each time point as the trace of the local Jacobian matrix $Tr(J)$. The Jacobian matrices are inferred by multiview distance regularised S-map (MDR S-map). The size of the Jacobian is fixed over time, as $s \times s$ (row \times column), where s is the number of network nodes in the food web (i.e. $s = 35$ for the food web in Lake Monona). In **d**, the temporal stability of the food web and of each trophic group are computed as the coefficient of variation of the total community abundance and each trophic group, respectively, using a time window of 1.5 years. Species asynchrony of the food web and of each trophic group is computed using a moving window. In **e**, time varying the species richness (i.e. the sampled species richness based purely on species presence and absence) and Simpson diversity are calculated from the data, and combined with time varying temperature data.

Reviewer comments, second round

Reviewer #1 (Remarks to the Author):

The authors apply empirical dynamic modeling (EDM) to determine effects of temperature and biodiversity on the structural stability and temporal stability of natural aquatic systems. They take advantage of the fact that there are long-term time series from 19 aquatic food webs, including lakes, rivers and coastal marine sites. The authors use convergent cross mapping to infer species interactions and S-maps to estimate interaction strengths through time. This allows both temporal and structural stability to be estimated from the constructed Jacobian, where recent theory, volume contraction rate, is used to estimate the structural stability from the trace of the Jacobian.

Comments

I reviewed an earlier version of this manuscript. The authors have addressed comments on the earlier version of the manuscript have been addressed and the responses seem satisfactory to me. However, there is an area where the manuscript can be improved, which is the biological interpretation of the results of the mathematical analysis.

In the Discussion the authors offer some biological mechanisms for the effects of temperature and biodiversity on structural and temporal stability. These mechanisms perhaps can be elaborated further. For example, the authors note that "Higher mean temperature was associated with lower link density, which reduced temporal stability of food webs." That is perhaps expected from indirect effects of metabolic theory, which predicts that an increase in temperature has a greater effect on increasing attack rate than it does on lowering handling time (Petchey et al. 2010). This is explained by Petchey in terms of a lower activation energy in the Arrhenius function for handling rate than attacking rate, which causes differences in the sensitivity to temperature. As Petchey et al. notes, this should lead to greater specialization by predators (on prey with shorter handling times), and thus decreased connectance (link density) within a food web. This is reflected in data from studies cited by Petchey. See also Rall et al. (2010).

The authors also note that warmer temperature increased the synchrony of all species in the food web and that this increase was associated with decreased temporal stability. That could be associated with tighter trophic interactions predicted O'Connor (2009), at least for herbivore-autotroph interactions. As metabolic theory predicts herbivore feeding increases faster than autotroph photosynthesis with increasing temperature, this predicts stronger herbivore control of autotrophs. Though not mentioned by O'Connor, that could have some positive effect on synchrony.

Petchey, O.L., Brose, U. and Rall, B.C., 2010. Predicting the effects of temperature on food web connectance. *Philosophical Transactions of the Royal Society B: Biological Sciences*, 365(1549), pp.2081-2091. Rall, B.C., Vucic-Pestic, O.L.I.V.E.R.A., Ehnes, R.B., Emmerson, M. and Brose, U., 2010. Temperature, predator-prey interaction strength and population stability. *Global Change Biology*, 16(8), pp.2145-2157.

O'Connor, M.I., 2009. Warming strengthens an herbivore-plant interaction. *Ecology*, 90(2), pp.388-398.

Reviewer #2 (Remarks to the Author):

See document attached.

Reviewer #3 (Remarks to the Author):

The authors appear to have largely addressed all of my comments from my review, along with the

other reviewers, and I appreciate the effort they put in to revising this manuscript. I do not have any further issues.

Review - round 2

Zhao et al (2023) *Nature Communications*

Comments to Authors

I believe that the authors have done a great job in responding to my criticisms as well as from other reviewers. They have conducted many analyses to address the points I raised and I am very pleased with the new version of the manuscript. I only have a few minor suggestions to improve the manuscript a bit more before publication.

Minor points

- Line 54: “impacts”?
- Line 56: Maybe add “respectively”?
- Lines 64 and 72: “no information is available for” does not read very well. In general, check carefully the additions (blue sections) throughout the main text, I think that the writing could be slightly improved in some of those additions.
- Lines 151-153: I believe that the fact that effects are system-dependent is important and should be emphasized more. Although the authors included an explanation in the Discussion of how temperature may destabilize food webs, it is hard to believe that the effect will be the same for every food web analyzed. The authors could, for example, explicitly mention for how many food webs (out of 19) the effects of temperature, richness, and diversity were positive and negative.
- Line 159: “mostly driven”? I feel that the results of contribution from predators in Fig 3 are not that strong for the authors to claim that they explain the temperature effect.
- Line 208: “are possible explanations to”?
- Line 265: Perhaps remove “long-term” as I had suggested before for the title.
- Line 461: Remove “and”
- Line 483: Mention here if there were any missing points after taking the seasonal average. If yes, mention how you dealt with missing points.
- Line 523: Is this mean absolute error of simplex abundance predictions?
- Line 588: Define $x_i(t)$ and use same notation as used in MDR S-map equation.
- Line 625: I would recall at the end of this paragraph that each element in B approximates $\frac{\partial x_i(t+1)}{\partial x_j(t)}$.

REVIEWERS' COMMENTS

Responses to Reviewer #1

(The Line numbers below refer to the color highlighting version of manuscript)

Reviewer #1 (Remarks to the Author):

The authors apply empirical dynamic modeling (EDM) to determine effects of temperature and biodiversity on the structural stability and temporal stability of natural aquatic systems. They take advantage of the fact that there are long-term time series from 19 aquatic food webs, including lakes, rivers and coastal marine sites. The authors use convergent cross mapping to infer species interactions and S-maps to estimate interaction strengths through time. This allows both temporal and structural stability to be estimated from the constructed Jacobian, where recent theory, volume contraction rate, is used to estimate the structural stability from the trace of the Jacobian.

Comments

I reviewed an earlier version of this manuscript. The authors have addressed comments on the earlier version of the manuscript have been addressed and the responses seem satisfactory to me. However, there is an area where the manuscript can be improved, which is the biological interpretation of the results of the mathematical analysis.

In the Discussion the authors offer some biological mechanisms for the effects of temperature and biodiversity on structural and temporal stability. These mechanisms perhaps can be elaborated further. For example, the authors note that "Higher mean temperature was associated with lower link density, which reduced temporal stability of food webs." That is perhaps expected from indirect effects of metabolic theory, which predicts that an increase in temperature has a greater effect on increasing attack rate than it does on lowering handling time (Petchey et al. 2010). This is explained by Petchey in terms of a lower activation energy in the Arrhenius function for handling rate than attacking rate, which causes differences in the sensitivity to temperature. As Petchey et al. notes, this should lead to greater specialization by predators (on prey with shorter handling times), and thus decreased connectance (link density) within a food web. This is reflected in data from studies cited by Petchey. See also Rall et al. (2010).

The authors also note that warmer temperature increased the synchrony of all species in the food web and that this increase was associated with decreased temporal stability. That could be associated with tighter trophic interactions predicted O'Connor (2009), at least for herbivore-autotroph interactions. As metabolic theory predicts herbivore feeding increases faster than autotroph photosynthesis with increasing temperature, this predicts stronger herbivore control of autotrophs. Though not mentioned by O'Connor, that could have some positive effect on synchrony.

Petchey, O.L., Brose, U. and Rall, B.C., 2010. Predicting the effects of temperature on food web connectance. *Philosophical Transactions of the Royal Society B: Biological Sciences*, 365(1549), pp.2081-2091.

Rall, B.C., Vucic-Pestic, O.L.I.V.E.R.A., Ehnes, R.B., Emmerson, M. and Brose, U., 2010. Temperature, predator–prey interaction strength and population stability. *Global Change Biology*, 16(8), pp.2145-2157.

O'Connor, M.I., 2009. Warming strengthens an herbivore–plant interaction. *Ecology*, 90(2), pp.388-398.

Re1: Thanks again for your comments. In the revised manuscript, we have elaborated further these biological mechanisms, early in the Discussion section (lines 235-244). Specifically, we elaborated on the mechanisms of how temperature could reduce link density by increasing attack rate to a greater extent than decreasing handling time (lines 239-244). Similarly, the mechanisms of how temperature increased the synchrony of all species in the food web by tightening trophic interactions are now added in revised manuscript (lines 235-239). We hope those changes help readers understand our results better.

Responses to Reviewer #2

(The Line numbers below refer to the color highlighting version of manuscript)

Reviewer #2 (Remarks to the Author):

Comments to Authors

I believe that the authors have done a great job in responding to my criticisms as well as from other reviewers. They have conducted many analyses to address the points I raised and I am very pleased with the new version of the manuscript. I only have a few minor suggestions to improve the manuscript a bit more before publication.

Re2: Thank you very much for your constructive feedback. Following your suggestion, we have revised them as below.

Minor points

- Line 54: “impacts”?

Re3: In the revised manuscript, we have changed the term “impact” to “impacts” accordingly (line 52).

- Line 56: Maybe add “respectively”?

Re4: We have added “respectively” (line 54).

- Lines 64 and 72: “no information is available for” does not read very well. In general, check carefully the additions (blue sections) throughout the main text, I think that the writing could be slightly improved in some of those additions.

Re5: Thanks for pointing this out. We have revised these additions (lines 60-61, 68-69).

- Lines 151-153: I believe that the fact that effects are system-dependent is important and should be emphasized more. Although the authors included an explanation in the Discussion of how temperature may destabilize food webs, it is hard to believe that the effect will be the same for every food web analyzed. The authors could, for example, explicitly mention for how many food webs (out of 19) the effects of temperature, richness, and diversity were positive and negative.

Re6: Thanks for this constructive feedback. In the revised manuscript, we have explicitly mentioned for how many food webs the effects of temperature, richness, and diversity were positive or negative (lines 146-150). We mentioned the system-dependent effects in the Discussion section (lines 197-198).

- Line 159: “mostly driven”? I feel that the results of contribution from predators in Fig 3 are not that strong for the authors to claim that they explain the temperature effect.

Re7: We have changed this accordingly.

- Line 208: “are possible explanations to”?

Re8: We have changed this accordingly.

- Line 265: Perhaps remove “long-term” as I had suggested before for the title.

Re9: We have removed “long-term”.

- Line 461: Remove “and”

Re10: We have removed “and”.

- Line 483: Mention here if there were any missing points after taking the seasonal average. If yes, mention how you dealt with missing points.

Re11: We have two missing points across whole dataset (accounting for nearly 0.17% of whole data), after taking the seasonal average. Specifically, one missing point was from Narragansett Bay (winter of 1979), and the other one was from Wadden Sea (summer of 1989). Those 2 missing data points were linearly interpolated using the ‘*na.approx*’ function in the package of zoo, following (Karakoç et al. 2020). We added those details in the revised manuscript (lines 508-512).

Karakoç, C., Clark, A. T. & Chatzinotas, A. Diversity and coexistence are influenced by time-dependent species interactions in a predator–prey system. *Ecol. Lett.* **23**, 983–993 (2020).

- Line 523: Is this mean absolute error of simplex abundance predictions?

Re12: Thanks for pointing this out. The technique we used here is indeed “simplex projection”, which is same as Ushio *et al* (2018) and Karakoç *et al* (2020). In the revised manuscript, we have explicitly stated the technique we used, by rewriting the sentence as “*We used simplex projection to select the best E that gave the lowest mean absolute error*” (lines 547-548).

Ushio, M. et al. Fluctuating interaction network and time-varying stability of a natural fish community. *Nature* **554**, 360–363 (2018).

Karakoç, C., Clark, A. T. & Chatzinotas, A. Diversity and coexistence are influenced by time-dependent species interactions in a predator–prey system. *Ecol. Lett.* **23**, 983–993 (2020).

- Line 588: Define $x_i(t)$ and use same notation as used in MDR S-map equation.

Re13: In the revised manuscript, we have defined the $x_i(t)$ among $\frac{\partial x_i(t+1)}{\partial x_j(t)}$ (line 606). Since according to the journal format, an italic letter (i.e. $x_i(t)$ in line 606 represents a single value, and a bold letter (i.e. $\mathbf{X}(t+1)$ in line 624 in equation (1) represents a vector of values, we had to use separate notation for $x_i(t)$ and $\mathbf{X}(t+1)$.

- Line 625: I would recall at the end of this paragraph that each element in B approximates $\frac{\partial x_i(t+1)}{\partial x_j(t)}$.

Re14: we have added it accordingly (line 644).

Responses to Reviewer #3

Reviewer #3 (Remarks to the Author):

The authors appear to have largely addressed all of my comments from my review, along with the other reviewers, and I appreciate the effort they put in to revising this manuscript. I do not have any further issues.

Re15: Thank you very much for all your comments.